# Improving constraint-based discovery with robust propagation and reliable LLM priors

## Abstract

Learning causal structure from observational data is central to scientific modeling and decision-making. Constraint-based methods aim to recover conditional independence (CI) relations in a causal directed acyclic graph (DAG). Classical approaches such as PC and subsequent methods orient v-structures first and then propagate edge directions from these seeds, assuming perfect CI tests and exhaustive search of separating subsets—assumptions often violated in practice, leading to cascading errors in the final graph. Recent work has explored using large language models (LLMs) as experts, prompting sets of nodes for edge directions, and could augment edge orientation when assumptions are not met. However, such methods implicitly assume perfect experts or predictable error rates, which is unrealistic for hallucination-prone and unstable LLMs. We propose MosaCD, a causal discovery method that propagates edges from a high-confidence set of seeds derived from both CI tests and LLM annotations. To filter hallucinations, we introduce shuffled queries that exploit LLMs' positional bias, retaining only high-confidence seeds. We then apply a novel confidence-down propagation strategy that orients the most reliable edges first, and can be integrated with any skeleton-based discovery method. Across multiple real-world graphs, MosaCD achieves higher accuracy in final graph construction than existing constraint-based methods, largely due to the improved reliability of initial seeds and robust propagation strategies.

## 1 Introduction

Causal discovery methods aim to recover a graph describing cause-effect relationships among a set of variables. One prominent family—originating with the famed PC algorithm (Spirtes et al., 2000)—are constraint-based methods. These methods conduct a series of conditional independence (CI) tests to rule out edges in the graph, then orient the remaining edges. Orientation begins with a "seed" set of edges determined by v-structures and proceeds by propagation rules that iteratively orient additional edges. Constraint-based methods are widely-used in practice for their theoretical guarantees, flexibility across data types, and interpretability of outputs (Spirtes et al., 2000). However, each step is prone to error accumulation. In theory, CI tests must perfectly distinguish dependence from independence (Spirtes et al., 2000) across all conditioning subsets (Kalisch & Bühlman, 2007), yet in practice CI tests with finite samples are noisy and exhaustive subset search is infeasible. During the edges orientation phase, these errors can be amplified: in order for seed edge directions to be correctly determined, the algorithm must correctly determine conditional dependencies across many different subsets of nodes (e.g., orienting a v-structure requires proving a node never appears in any separating set for a given pair). Nevertheless, constraint-based methods remain reliant on v-structure orientation, as no better alternative exists for determining an initial seed set of oriented edges.

Recent advances in LLMs offer new opportunities for seeding edges orientations. Despite their imperfections, LLMs contain broad knowledge that can be used to infer pairwise causal relationships (Kiciman et al., 2023; Vashishtha et al., 2025). Existing work has explored prompting LLMs with causal queries (e.g., "Does $A$ cause $B$?" (Kiciman et al., 2023; Vashishtha et al., 2023); "Is $A$ conditionally independent of $B$ given $C, D, \ldots$?" (Cohrs et al., 2024)), or using LLM information as a causal order prior (Vashishtha et al., 2025) or constraint to be enforced (Hasan & Gani, 2023). However, existing work combining LLMs with causal discovery algorithms keeps the two components

entirely separate, either querying the LLM first and feeding the results into an existing algorithm as priors (Hasan & Gani, 2023), or running a standard algorithm like PC and then using the LLM for post hoc orientation (Vashishtha et al., 2025; Khatibi et al., 2024). In both case, the discovery algorithm itself remains unchanged.

We study how causal discovery algorithms can be themselves redesigned to take advantage of LLMs as a complementary source of seed information (though our methods are also applicable to other sources of information like human experts). Our main contribution is a new causal discovery algorithm, MosaCD, which is designed to capitalize on this resource. MosaCD constructs a high-confidence set of seed edges using both CI test results and LLM annotations. Empirically, this set of seeds yields far fewer false positives than existing algorithms, reducing error cascades. We further introduce new propagation rules tailored to prioritize orientations supported by more reliable evidence. A central element of MosaCD is a prompting strategy that mitigates hallucinations. The key observation is that false positives due to hallucinations or overconfidence are uniquely destructive in the causal discovery process because they cause cascading errors during propagation—the overall algorithm will perform better if low-confidence edges are simply left un-oriented. We implement a simple but effective filtering strategy which exploits LLMs' tendency to select the first multiple-choice option when the true answer is unknown. To this end, we design shuffled queries that exploit LLMs' positional bias: orientations are randomized across multiple-choice orderings, and only consistently chosen orientations are retained as seeds. We evaluate MosaCD on 10 causal discovery benchmark datasets of up to 76 nodes and reach new state-of-the-art performance for constraint-based discovery, driven by more reliable seeding and robust propagation.

Our contributions are:

1. We propose MosaCD, a constraint-based causal discovery algorithm that combines CI-based orientation with robust LLM-based seeding and a confidence-prioritized propagation procedure.

2. We demonstrate how LLMs can be adapted for robust seeding in causal discovery with a domain-specific querying strategy to minimize hallucination influence.

3. We evaluate MosaCD on 10 real-world datasets and and show strong, consistent performance, particularly in information-heavy domains.

## 2 RELATED WORK

**Constraint-based causal discovery** Constraint-based learning of causal structure involves inferring edges and orientations with CI relations and logical rules with a sequential algorithm. The original PC algorithm (Spirtes et al., 2000) infers a causal graph by removing edges via CI tests and orienting the remaining ones with logical rules, assuming no hidden confounders. FCI (Spirtes et al., 2013) generalizes PC to allow for latent confounders and selection bias, returning MAPs rather than completed partially directed acyclic graphs (CPDAGs). PC-stable (Colombo et al., 2014) addresses PC's variable order dependency by fixing adjacency sets across each conditioning set size. To limit the influence of false colliders (from v-structure orientations), Conservative PC (CPC) (Ramsey et al., 2012) requires unanimity among observed separating sets to orient an edge, still assuming the separating sets are comprehensive. Post-hoc consistency procedures revisit the CI test results to reconcile a partially directed acyclic graph (PDAG) with the skeleton's evidence. PC-max (Ramsey, 2016) focuses on conditioning sets with the most significant p-values to avoid contradictions. These methods still fundamentally rely on v-structure orientation for seeding initial orientations, propagating the rest of the graph from these assumed-correct edges by not creating new v-structures. We focus on constraint-based discovery, although we acknowledge score-based methods that aim to learn an entire optimal graph structure such as NOTEARS (Zheng et al., 2018), DAG-GNN (Yu et al., 2019a), or GES (Chickering, 2002) will have different strengths and weaknesses.

**Incorporating domain knowledge to causal discovery** Constraint-based methods offer the flexibility of easily adding domain constraints or priors, as opposed to score-based methods where domain knowledge has to be tied into the global objective. Tiered orders and path constraints can prune orientations after PC (or variants), then be closed under orientation rules (Meek, 2013). While scalable, this still inherits the v-structure-first bias as knowledge is applied after initial orientations. Beyond this paradigm, Hyttinen et al. (2014) encodes tested (in)dependences together with prior knowledge

as logical constraints and minimizes the total weight of violated constraints, which inherits NP-hard worst-case complexity and degrades with dense graphs. Claassen & Heskes (2012) assigns Bayesian reliabilities to (in)dependence claims and processes them in decreasing reliability, returning a single model with a confidence tag per decision, but it requires enumerating (parts of) the Markov equivalence class, which is also computationally expensive.

**LLMs for causal discovery** LLMs have been shown to have relevant domain knowledge valuable for causal discovery. However, it is difficult to tell when an LLM is accurately using this information, or does not know the answer and is simply providing a response. Some methods aim to construct a set of constraints (or even the entire graph) using LLMs in a questionnaire style, essentially asking "Does $A$ cause $B$?" (Kiciman et al., 2023; Vashishtha et al., 2023; Jiralerspong et al., 2024), or "Is $A$ conditionally independent of $B$ given $\{C, D, \ldots\}$?" (Cohrs et al., 2024), or using LLM information as a prior selector (Vashishtha et al., 2025; Havrilla et al., 2025; Long et al., 2023; Ban et al., 2025) or constraint (Takayama et al., 2024). However, these methods implicitly assume that direct LLM outputs are reliable or have predictable error rates, without accounting for well-documented issues such as hallucination and positional bias (see below). MosaCD instead relies on a domain-specific querying procedure with shuffling and voting that can accurately filter out LLM hallucinations.

**Hallucination detection in LLMs** LLMs often refuse to acknowledge uncertainty. Given a multiple-choice question, they may just select the first option that is not "I don't know" if they are unsure. Empirical studies report positional and presentation biases and a reluctance to admit uncertainty; simple shuffle-and-vote mitigations help but do not absolve the need for further calibration (Wang et al., 2023; Pezeshkpour & Hruschka, 2023). Existing work addressing hallucinations typically involves either direct access to the LLM (Farquhar et al., 2024), or ability to fine-tune (Zhang et al., 2024). Cheaper methods that work with prompts only use self-reflection prompting strategies (Manakul et al., 2023) or LLM-generated confidence scores (Zhao et al., 2024), which requires an LLM to reason about when it is wrong. A cheap, prompt-only method for hallucination filtering that can be applied in scale without needing calibration would benefit causal discovery methods looking to extract priors from an LLM.

## 3 PRELIMINARIES

**Notations.** Let $G = (V, E)$ be the (unknown) ground-truth DAG, where $V$ is the set of observed variables and $E$ is the edge set. For disjoint $X, Y, S \subseteq V$, write $X \perp Y \mid S$ if $S$ $d$-separates $X$ and $Y$ in $G$, and $X \not\perp Y \mid S$ otherwise. We use $X - Y$ for an undirected edge, $X \to Y$ for a directed edge, and $X \rightsquigarrow Y$ for a (semi-)directed path i.e., a path from $X$ to $Y$ in which all arrows, if present, point forward from $X$ toward $Y$). We use curly braces $\{X, Y, \ldots\}$ to denote an unordered node set. A triple $X - Z - Y$ is *unshielded* if $X$ and $Y$ are non-adjacent but both are adjacent to $Z$. A partially directed acyclic graph (PDAG) is an acyclic graph whose edges may be directed or undirected.

**PC-style skeleton search (PC, CPC, PC-stable).** The procedure (Spirtes et al., 2000) starts from the complete undirected graph $K_V$ and removes an unordered edge between $X$ and $Y$ whenever a conditional independence (CI) test accepts $X \perp Y \mid S$ for some $S \subseteq V \setminus \{X, Y\}$. The resulting undirected graph is $\widehat{Skel}_\Sigma = (V, \hat{E})$. Along the way, we maintain a minimal sepset record $\Sigma$, where for each nonedge $\{X, Y\} \notin \hat{E}$, the set $\Sigma(X, Y) = \Sigma(Y, X) \subseteq 2^{V \setminus \{X, Y\}}$ collects all conditioning sets $S$ for which $X \perp Y \mid S$ was accepted (e.g., based on a p-value threshold). PC and PC-stable typically record only one separating set per nonedge, while CPC records multiple.

**Intuition of MosaCD.** Given the skeleton, traditional PC algorithms require an initial set of seed orientations to enable further propagation, e.g., via Meek's rules (Meek, 2013). Colliders serve as the seeds (also called "v-structure orientation"): for each unshielded triple $X - Z - Y$, if $Z \notin S$ for all $S \in \Sigma(X, Y)$, then $Z$ is oriented as a collider $X \to Z \leftarrow Y$. The intuition is that if $Z$ never appears in a separating set, then the alternative non-collider configurations $X \leftarrow Z \to Y$, $X \to Z \to Y$, $X \leftarrow Z \leftarrow Y$ are ruled out, leaving only the collider. In practice, however, CI tests are noisy and statistically asymmetric: a small p-value provides strong evidence for dependence, but a large p-value may reflect limited power rather than genuine independence. Thus, identifying a *collider* is less robust than identifying a *non-collider*: if $Z \notin S$ for any $S \in \Sigma(X, Y)$, this absence could be due to low power rather than $Z$ being a collider, whereas if $Z \in S$ for some $S \in \Sigma(X, Y)$, it provides strong evidence that $Z$ is a non-collider. While CI tests can be reliable

for coarse use in adjacency discovery (removing an edge takes just one CI test), their fine-grained use in collider orientation is much more fragile (orienting colliders requires all relevant CI tests be accurate). Motivated by this observation, we propose (i) replacing collider-based seeding with LLM-based orientation seeding, and (ii) prioritizing identifying non-colliders over colliders. This enables orientations that traditional PC algorithms cannot infer: in particular, even when $Z$ is identified as a non-collider, PC alone cannot resolve the orientation among the non-collider configurations without additional seeds.

## 4  METHOD

MosaCD takes as input a dataset $\mathcal{D}$, the corresponding variables $V$ with names and descriptions, and an LLM, and outputs a fully oriented DAG (Algorithm 1). MosaCD consists of 5 steps. First, it constructs the undirected skeleton using a constraint-based method (e.g., PC, CPC, PC-stable), yielding $G_{\text{skel}}$ and a minimal sepset record $\Sigma$ with CI p-values. Second, it uses an LLM to generate a set of high-confidence seed orientations, supplying variable names/descriptions and $\Sigma(X, Y)$, reducing LLM positional bias and hallucination by shuffling the answer order and repeating. This is more robust than collider-based seeding, which is sensitive to CI test inaccuracy, limited power and the order of processing. Third, we propagate orientations iteratively, where for unshielded triples $X - Z - Y$, we prioritize *non-collider* evidence ($Z$ in *all* minimal sepsets of $\Sigma(X, Y)$) over collider evidence ($Z$ in *none* of the minimal sepsets of $\Sigma(X, Y)$), as the latter may instead reflect limited power. Fourth, MosaCD resolves the remaining undirected edges by selecting the orientation that yields the fewest conflicts with $\Sigma$ with ties remaining undirected. Fifth, optionally, leftover undirected edges can be oriented using a topological order derived from aggregated LLM votes in Step 2.

---

**Algorithm 1** MosaCD

**Input:** Dataset $\mathcal{D}$ with variables $V$ (names and descriptions)
1: **Skeleton search:** $(G_{\text{skel}}, \Sigma) \leftarrow \text{SKELSEARCH}(\mathcal{D})$ and initialize PDAG $P \leftarrow G_{\text{skel}}$  ▷ PC/CPC/PC-stable
2: **LLM-based orientation seeding:** Query the LLM to propose high-confidence directions for undirected edges in $P$ using variable names, descriptions, and $\Sigma(X, Y)$ (minimal sepsets and CI p-values)  ▷ Shuffle answer order and repeat queries to reduce positional bias & hallucination
3: **Repeat until $P$ converges**
4:     **Repeat until $P$ converges**                                                            ▷ Rule closure
5:         **Unsupervised propagation (Meek R2, generalized):** If there exists a directed path $X \rightsquigarrow Y$ in $P$, and $X - Y$, orient $X \rightarrow Y$.
6:         **CI-supervised propagation:** Sort unshielded partially ordered triples $X \rightarrow Z - Y$ by descending $\max p$ in $\Sigma(X, Y)$; orient $Z \rightarrow Y$ if $Z$ is in all minimal sepsets of $\Sigma(X, Y)$; orient $Y \rightarrow Z$ if $Z$ is in none
7:         **Collider orientation:** Sort unshielded unordered triples $X - Z - Y$ by descending $\max p$ in $\Sigma(X, Y)$. Orient $X \rightarrow Z \leftarrow Y$ if $Z$ is in none of the minimal sepsets in $\Sigma(X, Y)$
8:     **Least-conflict orientation:** For each undirected $X - Y$ (in random order), choose the direction with the fewest conflicts w.r.t. $\Sigma$; leave undirected on ties
9: **(Optional) Final orientation via votes:** For remaining undirected edges, orient using previous LLM votes
10: **return** $P$

---

**Step 1: Skeleton search.** MosaCD initializes a skeleton $G_{\text{skel}}$ and sepset record $\Sigma$ from dataset $\mathcal{D}$ using a constraint-based algorithm such as PC, CPC, or PC-stable. The PDAG $P$ is initialized as $G_{\text{skel}}$. For each conditionally independent pair $\{X, Y\}$, $\Sigma(X, Y)$ contains at least one sepset together with the corresponding CI p-value. Throughout, $\Sigma(X, Y)$ denotes the collection of *minimal* separating sets recorded by the skeleton procedure.

**Step 2: LLM-based orientation seeding.** We generate a set of high-confidence seed orientations for undirected edges by querying an LLM. For each undirected edge $X - Y$ (in arbitrary order), we provide the LLM with variable names, variable descriptions, and $\Sigma(X, Y)$ including minimal sepsets and CI p-values. To reduce positional bias and hallucination, we randomize the order of candidate answers (e.g., both "$X \rightarrow Y$ or $Y \rightarrow X$" and the reverse) and repeat each query 5 times. Edges with consistent answers (being the majority vote in both orders) are retained as initial seeds. We discard any proposed seed $X \rightarrow Y$ that (i) contradicts $\Sigma$ at any unshielded triple, or (ii) would create a directed or semi-directed cycle in $P$.

**Step 3: Iterative orientation propagation.** MosaCD repeats the following steps until convergence.

3.1 **Unsupervised acyclic propagation:** Meek R2: If $X - Y$ and $X \rightsquigarrow Y$ in $P$, set $X \rightarrow Y$.

3.2 **CI-supervised propagation:** For each unshielded and partially ordered triple $X \rightarrow Z - Y$, sort in descending order by $\max p$ in $\Sigma(X, Y)$, since larger p-values provide stronger evidence for conditional independence. Orient $Z \rightarrow Y$ if $Z$ appears in all saved minimal sepsets of $\Sigma(X, Y)$, and orient $Y \rightarrow Z$ if $Z$ appears in none. Prioritizing by $\max p$ ensures that orientations with stronger CI support are applied first.

3.3 **Collider orientation:** For each unshielded and unordered triple $X - Z - Y$, again sort by descending $\max p$ in $\Sigma(X, Y)$. Orient $X \rightarrow Z \leftarrow Y$ if $Z$ appears in none of the minimal sepsets in $\Sigma(X, Y)$.

**Step 4: Least-conflict orientation.** MosaCD resolves any remaining undirected edges by choosing the direction that conflicts least with the recorded conditional independences in $\Sigma$. For each undirected pair $X - Y$ (in random order), consider both $X \rightarrow Y$ and $Y \rightarrow X$. For each option, close the graph under the usual orientation rules and count how many statements in $\Sigma$ would be contradicted; pick the option with the smaller count. If the counts tie, leave $X - Y$ undirected. Steps 3-4 are repeated until nothing changes.

*Example.* After Step 3, suppose $U \rightarrow Y, V \rightarrow Y, W \rightarrow Y, X - Y$ is undirected, and $\Sigma$ contains $X \perp U \mid \{Y\}, X \perp V \mid \{Y\}, X \perp W$. Then $X \rightarrow Y$ opens the colliders $X \rightarrow Y \leftarrow W$ and $X \rightarrow Y \leftarrow V$ (2 conflicts), whereas $Y \rightarrow X$ makes $T \rightarrow Y \rightarrow X$ induce $X \not\perp W$ (1 conflict); by step 4, we choose $Y \rightarrow X$.

**(Optional) Step 5: Final orientation via votes.** If some edges remain undirected, we further use the LLM votes from Step 2 to complete the DAG. This is analogous to Vashishtha et al. (2025), but our LLM procedure additionally integrates shuffled answer orders to mitigate positional bias. Votes are aggregated into a weighted directed graph, and the weakest edges (least net support between two directions) are removed to break cycles. A topological order is then derived from this weighted digraph, and any remaining undirected edges are oriented according to this order, yielding the final DAG.

## 5 THEORETICAL ANALYSIS

We start by verifying the correctness of MosaCD's novel propagation strategy, showing that it recovers a PDAG consistent with the true DAG under idealized assumptions similar to those used to prove correctness of existing causal discovery algorithms. Specifically, we assume a perfect CI oracle and a seeding oracle that never returns answers inconsistent with the true graph (though it may abstain from answering), alongside the standard PC assumptions of causal sufficiency and that the observational distribution is Markov and adjacency-faithful to G. Although unrealistic, these assumptions establish that our propagation rules are correct in the same sense as prior methods: given correct inputs, they recover the unique PDAG consistent with the ground truth. We then examine departures from these assumptions, particularly the noisiness in CI tests that motivates MosaCD. To complement our empirical results, we provide a theoretical analysis in a stylized model, demonstrating that orienting non-colliders first (as MosaCD does) yields fewer errors than orienting colliders first (as in PC algorithms).

### 5.1 CORRECTNESS

We show that the orientation procedure in MosaCD returns the completed partially directed acyclic graph (CPDAG) of the ground-truth DAG $G$. The CPDAG of a DAG $G$ is the unique PDAG representing the Markov equivalence class of $G$: (i) it has the same skeleton and v-structures as $G$; (ii) a directed edge $X \rightarrow Y$ appears in the CPDAG iff it is compelled (i.e., oriented identically in every DAG in the equivalence class); and (iii) an undirected edge $X - Y$ appears in the CPDAG iff it is reversible (i.e., can be oriented in either direction within the class) (Andersson et al., 1997, Theorem 4.1). $\Sigma(X, Y)$ stores minimal separators (as produced by PC/PC-stable/CPC under a perfect oracle).

**Theorem 5.1.** *For any distinct nodes $X, Y \in V$ and any conditioning set $S \subseteq V \setminus \{X, Y\}$, assume: (i) **Causal Markov condition:** if $S$ d-separates $X$ and $Y$ in $G$, then $X \perp Y \mid S$ in the distribution; (ii) **Adjacency-Faithfulness:** if $X$ and $Y$ are adjacent in $G$, then $X \not\perp Y \mid S$ for any $S \subseteq V \setminus \{X, Y\}$; (iii) **Perfect CI oracle:** the CI oracle returns whether $X \perp Y \mid S$ in the distribution*

*induced by $G$ without error; (iv) **Skeleton consistency:** $\widehat{Skel}_\Sigma = Skel(G)$; and (v) **Correct seeds:** the initial seed set $E_{seed}$ is $\Sigma$-consistent (no arrowhead contradicts $\Sigma$) and acyclic (no directed or semi-directed cycles). Then running Step 3 of MosaCD until convergence returns the CPDAG of $G$, and Step 4 performs no additional orientations. Furthermore, when $E_{seed} = \emptyset$, Step 3 returns the same PDAG as PC, PC-stable, and CPC.*

See Appendix I for the proof.

**Remark 5.2.** *In Theorem 5.1, conditions (i)-(ii) are standard and ensure that the DAG is consistent with the underlying distribution (Spirtes et al., 2000). (iii)-(v) assume a correct initialization (skeleton, $\Sigma$, and seeds). Under these assumptions, MosaCD's orientation propagation is provably correct and coincides with existing PC algorithms in the absence of seeds, justifying its design.*

### 5.2 PRIORITIZING NON-COLLIDER OVER COLLIDER IDENTIFICATION IMPROVES ACCURACY

For an unshielded triple $X - Z - Y$, traditional PC-style algorithms prioritize identifying colliders by checking whether $Z$ is absent from the separation sets $\Sigma(X, Y)$. In contrast, our method prioritizes non-colliders by checking whether $Z$ is present in $\Sigma(X, Y)$. To illustrate the difference between these strategies, we analyze a stylized model of the search over conditioning sets, focusing on a setting in which errors from the CI test are independent across queries and the graph is sparse. We find that when CI tests are noisy—incurring both false positives and false negatives—prioritizing non-colliders yields higher accuracy.

**Level-wise search and error events.** Let $\ell = |C|$ denote the conditioning-set size. The search proceeds by levels $\ell = 0, 1, 2, \ldots$, testing $X \perp Y \mid C$ over all $C \subseteq V \setminus \{X, Y\}$ with $|C| = \ell$. At level $\ell$, if any candidate $C$ is accepted as a sepset, the search stops; we declare $Z$ a collider if $Z \notin C$, and a non-collider if $Z \in C$. Accordingly, a collider error occurs if $Z$ is a non-collider but $Z \notin C$, and a non-collider error occurs if $Z$ is a collider but $Z \in C$. We measure their relative frequency via

$$\mathcal{R}_\ell := \frac{\Pr(\text{collider error at level } \ell)}{\Pr(\text{non-collider error at level } \ell)}. \tag{1}$$

At level $\ell$, PC uses the **first** accepted sepset $C$ for $(X, Y)$. PC-stable has the same collider/non-collider decision as PC, but with adjacency sets frozen within level $\ell$ (order-invariant), so $\mathcal{R}_\ell$ matches PC. At the first level $\ell$ where independence holds, CPC gathers all minimal sepsets for $(X, Y)$ and orient the collider iff $Z$ is in **none**, treat as non-collider iff $Z$ is in **all**, otherwise leave the triple unoriented.

We compute $\mathcal{R}_\ell$ for the PC (the same as that of PC-stable) and CPC rules (denoted $\mathcal{R}_\ell^{\text{PC}}$ and $\mathcal{R}_\ell^{\text{CPC}}$). PC-stable has the same $\mathcal{R}_\ell$ as PC, since it only removes within-level order dependence. Whenever $\mathcal{R}_\ell > 1$ for a given rule set, tests of non-colliders (prioritized by MosaCD) will have a lower error rate than test of colliders (prioritized by existing algorithms). We make the following assumptions to make theoretical analysis tractable, noting they are not required for running MosaCD in practice.

**Assumption 5.3.** *(Simple CI test model) Conditional independence (CI) tests act independently across candidates/levels given truth labels, with false positive rate $\alpha$ and false negative rate $\beta$ that do not vary with $\ell$ or $C$. Thus, a true sepset rejects dependence with probability $1 - \beta$, while a non-sepset does so with probability $\alpha$.*

**Assumption 5.4.** *(Z controls the $X - Y$ path) (a) All $X - Y$ paths of length at most $2\ell + 1$ pass through $Z$; (b) whether the $X - Y$ path is open is determined fully by whether $Z$ is conditioned on.*

While these assumptions are deliberately simplified, they are designed to illustrate the core dynamic by isolating the impact of the single node $Z$ (intuitively, Assumption 5.4 describes a locally sparse graph without redundant $X - Y$ paths) and imposing a single set of parameters describing the performance of the CI test.

In this model, we obtain exact analytical expressions for $R_\ell^{\text{PC}}$ and $R_\ell^{\text{CPC}}$, derived and shown in the appendix. These expressions involve a number of combinatorial quantities, but in the asymptotic regime where the error rates of the CI test are small relative to the graph size (a necessary condition for the algorithm to not be overwhelmed with errors), we can further simplify and show that $R_\ell^{\text{PC}}$ and $R_\ell^{\text{CPC}}$ must be strictly above 1.

**Theorem 5.5.** *Let* $M = |V \setminus \{X, Y\}|$. *Suppose that* $\alpha, \beta = o\left(\frac{1}{M}\right)$ *and* $\ell = \Theta(1)$. *For* $M$ *sufficiently large compared to* $\ell$, *the error ratios satisfy*

$$R_\ell^{\text{CPC}} = \beta^{\binom{M-1}{\ell-1} - \binom{M-1}{\ell}} \cdot \frac{M - \ell}{\ell} + o\left(\frac{1}{M}\right) > 1$$

$$R_\ell^{\text{PC}} = \left(\frac{M}{M - \ell}\right)^2 (1 - o(1)) + o\left(\frac{1}{M^2}\right) > 1$$

*so that for both algorithms, the error rate among colliders will be higher than noncolliders.*

See Appendix J for the proof. The intuition is that there are more candidate subsets that do *not* contain $Z$ than subsets that do contain $Z$, so collider-first strategies have more opportunities to make a mistake. MosaCD flips the ordering, starting with candidates that do contain $Z$, since this set is smaller and the total number number of mistakes made in early stages will be limited. Numerical experiments (Appendix K) also show substantially lower FPRs under the non-collider first strategy across three standard skeleton learners, consistent with the theory.

## 6 EXPERIMENTAL RESULTS

We consider 10 benchmark datasets from the BNLearn repository (Scutari & Denis, 2014): Cancer, Asia, Child, Insurance, Water, Mildew, Alarm, Hailfinder, Hepar2, and Win95pts. These datasets range from 5 to 76 nodes and include both real and simulated graphs. For simulated datasets, we generate 20,000 samples each. We measure performance using the F1 score and structural Hamming distance (SHD) for detecting true edge orientations.

**Baselines.** We consider 3 skeleton search methods: PC (Spirtes et al., 2000), PC-stable (Colombo et al., 2014), and CPC (Ramsey et al., 2012), which are compatible with all downstream orientation strategies. Given a skeleton, we apply 13 baselines: (i) **PC**, which follows the standard procedure of the corresponding skeleton method to orient edges (PC, PC-stable, or CPC); (ii) **Meek** (Meek, 2013), which applies Meek's rules after PC to the skeleton to orient remaining edges (distinct from the original PC procedures); (iii) **Shapley-PC** (Russo & Toni, 2023), which orients edges using a Shapley-value-based feature importance procedure; (iv) **ILS-CSL** (Ban et al., 2023), an LLM-based method that incorporates statistical knowledge; v) **SCP** (Cohrs et al., 2024), another LLM-based orientation method. (vi) **Jiralerspong** (Jiralerspong et al., 2024), a pairwise LLM-based method that sees two nodes in each prompt; (vii) **Vashishtha** (Vashishtha et al., 2025), a pairwise LLM-based causal ordering method that sees the whole graph in each prompt; (viii) **Causal Disco** (Long et al., 2023), a method using LLM information as a prior that can handle erroneous information; (ix) **LLM-CD** (Ban et al., 2025) another method using noisy LLM information as a prior; -and commonly used score-based and continuous optimization methods applied independently of the skeleton (x) **NOTEARS** (Zheng et al., 2018), (xi) **DAG-GNN** (Yu et al., 2019b), (xii) **GES** (Chickering, 2002), (xiii) **Hill climbing (HC)** (Heckerman et al., 1995), each as implemented in gcastle (Zhang et al., 2021). These baselines represent both state-of-the-art LLM-based and non-LLM methods from constraint-based, score-based, and continuous-optimization fields. All methods are given access to the dataset, while LLM-based methods are additionally supplied with identical variable names and dataset metadata, and all use the same LLM backbone (GPT-4o-mini). Please see more details in Appendix L.

### 6.1 BENCHMARKING EXPERIMENTS

We applied MosaCD and baseline methods to 10 benchmark datasets. Results based on PC skeletons are reported in Table 1 and those based on PC-Stable and CPC skeletons are in Appendix A. We reached 2 main conclusions. First, MosaCD outperformed all baselines, achieving the best performance in 7 out of 10 datasets and the best performance overall (average MosaCD F1 score 0.81 vs. next best Vashishtha 0.69); MosaCD similarly outperformed baselines using PC-Stable and CPC skeletons (Appendix A). Similar conclusions are reached using the SHD as a metric (Appendix B). Second, MosaCD consistently outperformed other LLM-based methods (ILS-CSL, SCP, Jiralerspong, Vashishtha), suggesting that MosaCD makes better utility of available LLM knowledge. Similar results are observed when using SHD as a metric in Appendix B. We validate the

LLM's strong tendency towards positional bias in Appendix D and validate our improvements over constraint based methods are not primarily driven by memorization in Appendix E.

|  | Cancer (5) | Asia (8) | Child (20) | Insurance (27) | Water (32) |
|---|---|---|---|---|---|
| PC | 0.50 | 0.67 | 0.70 | 0.62 | 0.45 |
| Meek | 0.50 | 0.67 | 0.78 | 0.70 | 0.57 |
| Shapley-PC | **1.00** | 0.53 | 0.67 | 0.67 | 0.47 |
| ILS-CSL* | 0.50 | **0.93** | 0.83 | 0.70 | 0.60 |
| SCP* | 0.50 | 0.67 | 0.78 | 0.68 | 0.57 |
| Jiralerspong* | 0.57 | 0.55 | 0.10 | 0.21 | 0.00 |
| Vashishtha* | 0.75 | 0.93 | 0.67 | 0.70 | 0.65 |
| Causal Disco* | 0.57 | 0.86 | 0.56 | 0.34 | **0.87** |
| LLM-CD* | **1.00** | 0.80 | 0.40 | 0.29 | 0.24 |
| NOTEARS | 0.00 | 0.57 | 0.48 | 0.15 | 0.29 |
| DAG-GNN | 0.00 | 0.27 | 0.24 | 0.20 | 0.26 |
| GES | 0.50 | 0.12 | 0.17 | 0.26 | 0.29 |
| HC | 0.50 | 0.12 | 0.33 | 0.33 | 0.36 |
| MosaCD* (Ours) | **1.00** | **0.93** | **0.90** | **0.87** | 0.59 |
|  | Mildew (35) | Alarm (37) | Hailfinder (56) | Hepar2 (70) | Win95pts (76) |
| PC | 0.63 | 0.85 | 0.38 | 0.36 | 0.59 |
| Meek | 0.69 | 0.90 | 0.40 | 0.39 | 0.64 |
| Shapley-PC | 0.75 | 0.84 | 0.38 | 0.44 | 0.65 |
| ILS-CSL* | 0.89 | 0.85 | 0.44 | 0.54 | 0.69 |
| SCP* | 0.69 | 0.87 | 0.39 | 0.38 | 0.63 |
| Jiralerspong* | 0.07 | 0.09 | 0.10 | 0.16 | 0.03 |
| Vashishtha* | 0.78 | 0.72 | **0.60** | 0.66 | 0.74 |
| Causal Disco* | 0.61 | 0.51 | 0.33 | **0.81** | 0.55 |
| LLM-CD* | 0.24 | 0.30 | 0.22 | 0.32 | 0.17 |
| NOTEARS | 0.25 | 0.25 | 0.27 | 0.06 | 0.14 |
| DAG-GNN | 0.43 | 0.53 | 0.13 | 0.13 | 0.37 |
| GES | 0.23 | 0.26 | 0.33 | 0.45 | 0.51 |
| HC | 0.24 | 0.30 | 0.31 | 0.47 | 0.46 |
| MosaCD* (Ours) | **0.90** | **0.93** | 0.49 | 0.72 | **0.81** |

Table 1: **BNLearn evaluation.** F1 score for each dataset and method. Number of nodes is provided in the bracket. Best in **bold**, second best underlined. "*" denotes LLM-based methods.

We further evaluated the effectiveness of MosaCD's LLM-based orientation seeding (Step 2) compared to the standard PC procedure (orienting v-structures). Results are reported in Figure 1, averaged across using PC, PC-Stable, and CPC skeletons. First, MosaCD's seeding procedure identified substantially more true directions (avg MosaCD vs. PC true seed ratio 1.69) and markedly fewer false directions (avg 4.8% vs. 26.7%) than PC across the 10 datasets. MosaCD relied substantially less on collider orientation than PC (average MosaCD collider orientations 3.3 versus PC 28.5). Second, even in datasets with less informative variable descriptions (Hailfinder, Win95pts), MosaCD remained robust: it detected relatively few true directions but did not introduce excessive false seeds compared to PC (average 16.3% vs. 21.7% across such datasets), likely due to the hallucination filtering procedure. Full results are reported in Appendix C. Additional experiments varying sample size are reported in Appendix G and discussing runtime are in Appendix F.

## 6.2 SECONDARY ANALYSES AND ABLATION STUDIES

First, we assessed the robustness of MosaCD's LLM inference by replacing a proportion of variable descriptions in the "insurance" dataset with uninformative ones. Results are reported in Figure 2. While MosaCD's performance and the number of true orientation seeds declined expectedly as the proportion of uninformative variables increased, it consistently generated only a small number of false seeds and remained competitive or superior to the baseline, demonstrating robustness.

Second, we evaluated the robustness of MosaCD's propagation procedures (Steps 3-5) through ablation studies on the "Insurance" dataset. Specifically, we removed the LLM seeding step (Step

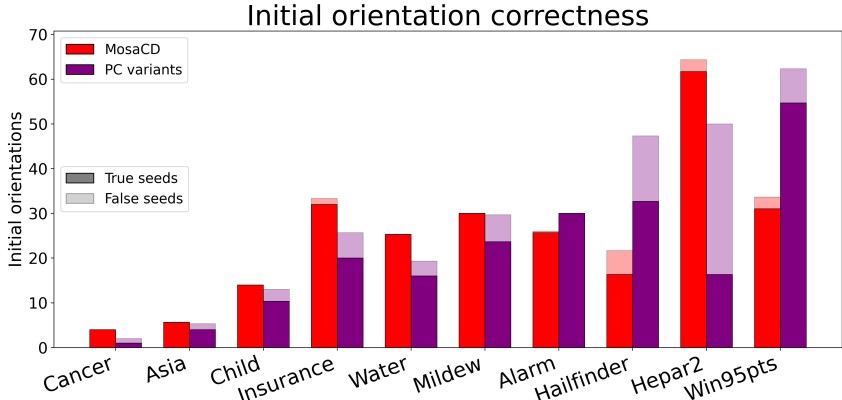

Figure 1: **Accuracy of orientation seeds.** Number of true and false directions discovered by MosaCD LLM-based orientation seeding (Step 2) and the standard PC procedure (orienting v-structures). Results are reported for in each dataset averaged across using PC, PC-stable, and CPC skeletons.

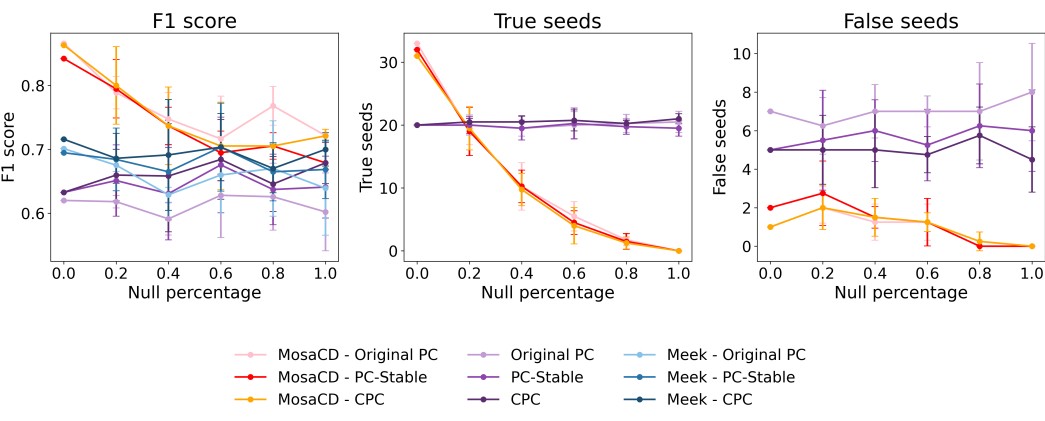

Figure 2: **Experiments with a proportion of uninformative variable descriptions.** F1 score, number of true seeds, and number of false seeds for MosaCD, PC, and Meek as the proportion of uninformative variable descriptions varies. Results are reported for using PC, PC-Stable, and CPC skeletons

2), varied the number of true seeds, and adjusted the proportion of false seeds, comparing against Meek's propagation rules. Results are reported in Figure 3. MosaCD consistently outperformed Meek both when varying the number of true seeds (with false seeds fixed at 0) and when varying the proportion of false seeds (with total seeds fixed at 20), demonstrating its effectiveness. We repeat this analysis in the Asia and Hepar2 datasets and achieve similar results (Appendix H).

Third, we varied the LLM backbone used in MosaCD, from efficient models (Claude-3.5-Haiku, GPT-4o-mini) to frontier reasoning models (GPT-5, Claude-Sonnet-4), as well as open-source models GPT-oss-120b, Llama 3 8B-instruct, and Qwen 3 32B, analyzing 3 representative datasets (Asia, Insurance, Hepar2). Results are reported in Figure 4. MosaCD maintained consistent performance across backbones, with the exception of GPT-oss-120b and Llama 3 8B-instruct, which exhibited slightly weaker results.

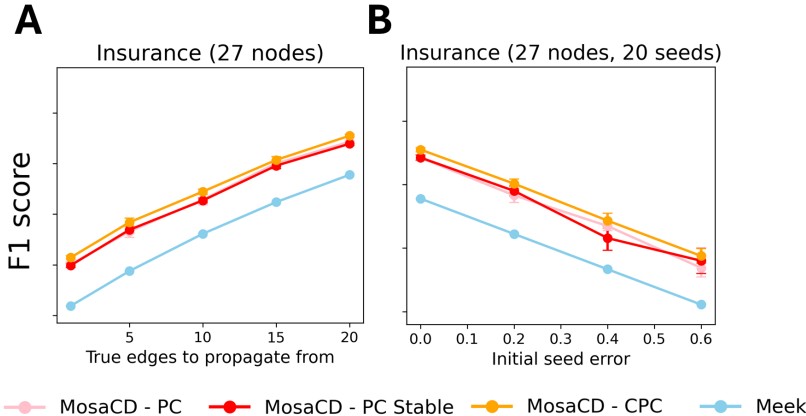

Figure 3: **Experiments varying number of true and false seeds.** F1 score for MosaCD (using PC, PC-Stable, and CPC skeletons) and Meek. **(A)** Varying the number of true seeds with false seeds fixed at 0. **(B)** Varying the proportion of false seeds with total seeds fixed at 20).

Figure 4: **MosaCD performance across LLMs.** F1 scores on the Asia, Insurance, and Hepar2 datasets across seven different LLMs (using original PC skeletons).

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

# A    RESULTS WITH DIFFERENT PC SKELETONS

|  | PC | Meek | Shapley-PC | SCP | MosaCD |
|---|---|---|---|---|---|
| Cancer (5 nodes) | 0.50 | 0.50 | 1.00 | 0.50 | 1.00 |
| Asia (8 nodes) | 0.75 | 0.93 | 0.53 | 0.93 | 0.93 |
| Child (20 nodes) | 0.90 | 0.90 | 0.67 | 0.90 | 0.90 |
| Insurance (27 nodes) | 0.65 | 0.74 | 0.73 | 0.72 | 0.86 |
| Water (32 nodes) | 0.47 | 0.57 | 0.47 | 0.59 | 0.63 |
| Mildew (35 nodes) | 0.64 | 0.71 | 0.74 | 0.71 | 0.87 |
| Alarm (37 nodes) | 0.85 | 0.90 | 0.84 | 0.87 | 0.93 |
| Hailfinder (56 nodes) | 0.42 | 0.43 | – | 0.44 | 0.47 |
| Hepar2 (70 nodes) | 0.42 | 0.44 | 0.45 | 0.43 | 0.72 |
| Win95pts (76 nodes) | 0.64 | 0.69 | 0.66 | 0.70 | 0.80 |

Table 2: **BNLearn evaluation (CPC).** F1 score using CPC's skeleton. "–" indicates method timed out after 12 hours.

|  | PC | Meek | Shapley-PC | SCP | MosaCD |
|---|---|---|---|---|---|
| Cancer (5 nodes) | 0.50 | 0.50 | 1.00 | 0.50 | 1.00 |
| Asia (8 nodes) | 0.67 | 0.67 | 0.53 | 0.67 | 0.93 |
| Child (20 nodes) | 0.70 | 0.78 | 0.67 | 0.78 | 0.86 |
| Insurance (27 nodes) | 0.65 | 0.72 | 0.73 | 0.69 | 0.84 |
| Water (32 nodes) | 0.48 | 0.58 | 0.47 | 0.59 | 0.63 |
| Mildew (35 nodes) | 0.69 | 0.73 | 0.74 | 0.75 | 0.87 |
| Alarm (37 nodes) | 0.85 | 0.90 | 0.84 | 0.87 | 0.96 |
| Hailfinder (56 nodes) | 0.39 | 0.41 | – | 0.39 | 0.57 |
| Hepar2 (70 nodes) | 0.40 | 0.43 | 0.47 | 0.42 | 0.71 |
| Win95pts (76 nodes) | 0.64 | 0.69 | 0.66 | 0.69 | 0.73 |

Table 3: **BNLearn evaluation (PC-Stable).** F1 score using PC-Stable's skeleton. "–" indicates indicates method timed out after 12 hours.

# B    RESULTS WITH SHD

|  | Cancer (5) | Asia (8) | Child (20) | Insurance (27) | Water (32) |
|---|---|---|---|---|---|
| PC | 2.0 | 3.0 | 10.5 | 28.0 | 53.0 |
| Meek | 2.0 | 3.0 | 6.0 | 19.0 | 38.0 |
| Shapley-PC | **0.0** | 4.0 | 19.0 | 27.5 | 52.5 |
| ILS-CSL* | 2.0 | **1.0** | 5.5 | 25.0 | 37.5 |
| SCP* | 2.0 | 3.0 | 6.0 | 21.0 | 38.0 |
| Jiralerspong* | 3.0 | 5.0 | 38.0 | 52.0 | 71.0 |
| Vashishtha* | 1.0 | **1.0** | 9.0 | 21.0 | 34.0 |
| Causal Disco* | 2.0 | 2.0 | 13.0 | 41.0 | **15.0** |
| LLM-CD* | **0.0** | 2.0 | 30.0 | 59.0 | 81.0 |
| DAG-GNN | 4.0 | 9.0 | 23.0 | 60.0 | 55.0 |
| GES | 2.0 | 9.0 | 63.0 | 105.0 | 71.0 |
| HC | 2.0 | 9.0 | 51.0 | 95.0 | 60.0 |
| MosaCD* (Ours) | **0.0** | **1.0** | **2.0** | **12.0** | 36.0 |
|  | Mildew (35) | Alarm (37) | Hailfinder (56) | Hepar2 (70) | Win95pts (76) |
| PC | 23.0 | 12.0 | 105.0 | 99.5 | 68.0 |
| Meek | 17.0 | 7.5 | 97.0 | 87.0 | 57.5 |
| Shapley-PC | 18.5 | 14.0 | 105.0 | 89.0 | 69.5 |
| ILS-CSL* | 10.0 | 12.0 | 96.5 | 77.5 | 54.0 |
| SCP* | 17.0 | 8.0 | 99.0 | 88.0 | 56.0 |
| Jiralerspong* | 50.0 | 80.0 | 72.0 | 137.0 | 135.0 |
| Vashishtha* | 13.0 | 15.0 | 77.0 | 59.0 | 46.0 |
| Causal Disco* | 24.0 | 27.0 | **53.0** | **36.0** | 69.0 |
| LLM-CD* | 52.0 | 59.0 | 92.0 | 117.0 | 160.0 |
| DAG-GNN | 50.0 | 35.0 | 77.0 | 119.0 | 93.0 |
| GES | 152.0 | 101.0 | 150.0 | 122.0 | 122.0 |
| HC | 140.0 | 92.0 | 162.0 | 119.0 | 133.0 |
| MosaCD* (Ours) | **8.0** | **6.0** | 90.0 | 52.0 | **37.0** |

Table 4: **BNLearn evaluation.** SHD for each dataset and method using the PC skeleton. Number of nodes is provided in the bracket. Best in **bold**. "*" denotes LLM-based methods.

## C  ORIENTATION CORRECTNESS BY SKELETON

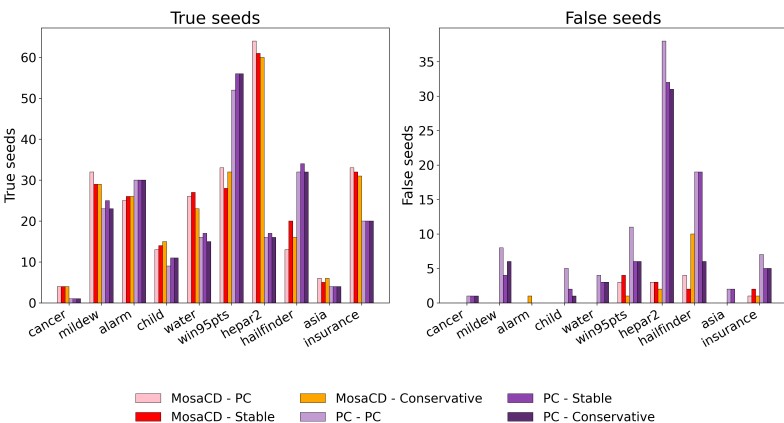

Figure 5: **Initial orientation correctness.** The average number of true and false seeds discovered by MosaCD and PC variants in each dataset, aggregated across PC, PC-stable, and CPC.

## D    POSITIONAL BIAS

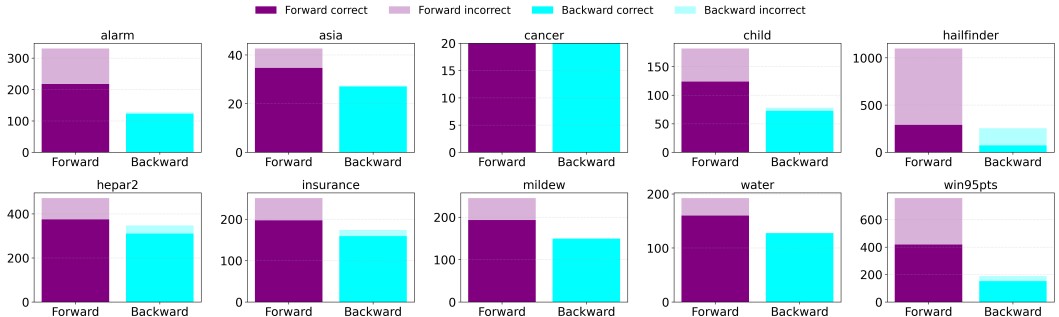

Figure 6: **Average count of total votes in each direction for a given prompt.** Forward appears before backward, and thus is chosen more frequently by the LLM. Shaded bars indicate votes in the wrong direction. Of note, I don't know was the first option, which was never selected.

# E    MEMORIZATION

We validate if MosaCD's improvements over constraint-based discovery are due to excessive LLM memorization of BNLearn datasets. We apply PC, Meek, and MosaCD to 2 datasets shown to have low memorization by modern LLMs (Srivastava et al., 2025). MosaCD achieves higher performance than both PC and Meek on both datasets, in the case of COVID Respiratory, much higher.

| Method | COVID Respiratory (11) | COVID Complications (63) |
|--------|------------------------|--------------------------|
| PC     | 0.31 | 0.24 |
| Meek   | 0.32 | 0.29 |
| MosaCD | 0.52 | 0.30 |

Table 5: F1 scores for each method across two less memorized datasets.

# F RUNTIME

Here we report the runtime in seconds for the skeleton search phase (step 1), LLM voting phase (step 2), and propagation phase (steps 3-5). We note that while costly, the LLM voting can be easily parallelized when using an API. Scaling to a larger dataset, Andes (223 nodes), takes 15 minutes of skeleton searching and 120 minutes of LLM voting.

| | Asia (8 nodes) | | | Insurance (27 nodes) | | | Hepar2 (72 nodes) | | |
|---|---|---|---|---|---|---|---|---|---|
| | PC | PC-Stable | CPC | PC | PC-Stable | CPC | PC | PC-Stable | CPC |
| Skeleton | 0.23 | 0.50 | 0.55 | 50.32 | 76.51 | 75.89 | 187.05 | 313.21 | 264.50 |
| LLM | 256.82 | 232.13 | 264.14 | 1501.94 | 1397.03 | 1434.65 | 2492.51 | 2453.28 | 2260.18 |
| Propagation | 0.00 | 0.00 | 0.00 | 0.01 | 0.01 | 0.01 | 0.03 | 0.03 | 0.03 |

Table 6: **Runtime analysis.** Runtime (seconds) of each stage of MosaCD for each dataset and PC variant.

# G    VARYING SAMPLE SIZE

## Varying sample size in Insurance

Figure 7: **Varying sample size.** Results showing F1 score, true seed counts, and false seed counts for MosaCD, PC, and Meek's rules for sample size from 100 to 10000.

## H    PROPAGATION ABLATIONS

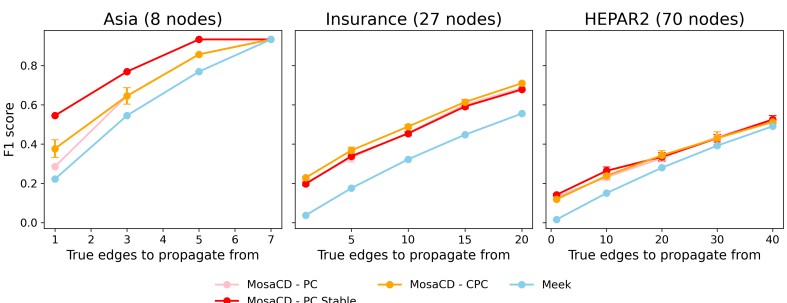

Figure 8: **Experiments varying number of true seeds.** F1 score for MosaCD (using PC, PC-Stable, and CPC skeletons) and Meek.

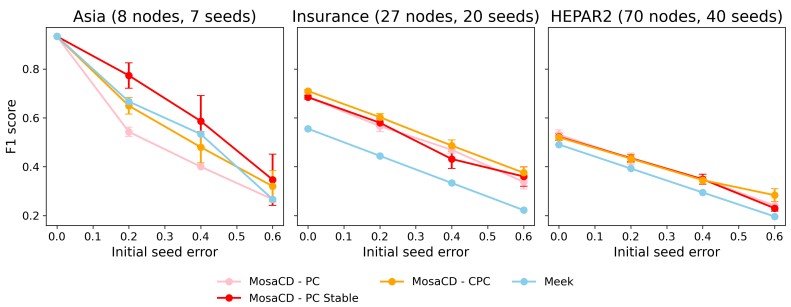

Figure 9: **Experiments varying number of false seeds.** F1 score for MosaCD (using PC, PC-Stable, and CPC skeletons) and Meek.

## I    PROOFS FOR 5.1

**Lemma I.1** (Collider/non-collider soundness w.r.t. $\Sigma$). *Consider an unshielded triple $X - Z - Y$ with $X$ nonadjacent to $Y$. Under a perfect CI oracle, exactly one of the following holds:*

    *1. $Z \notin S$ for all $S \in \Sigma(X, Y)$, which compels $X \to Z \leftarrow Y$;*

    *2. $Z \in S$ for all $S \in \Sigma(X, Y)$, which forbids a collider at $Z$.*

*Proof.* Standard separator consistency for unshielded triples implies exclusivity of $Z$ across minimal separators of $(X, Y)$; (a) and (b) are the two mutually exclusive cases, see Spirtes et al. (2000, Lemma 5.1.3). □

**Lemma I.2** (Confluent closure of the CI-guarded orientation phase). *Start from the PDAG on the true skeleton after inserting the seed arrows $E_{\text{seed}}$, assuming these seeds are $\Sigma$-consistent and introduce no directed or semi-directed cycles. With a perfect CI oracle, Step 3 terminates and returns a unique maximally oriented PDAG compatible with the skeleton and with $\Sigma \cup E_{\text{seed}}$. The result is independent of the order in which Steps 3.1-3.3 are applied.*

*Proof.* Define $\mathcal{G}_0$ as the PDAG obtained from the correct skeleton by adding the acyclic, $\Sigma$-consistent seeds $E_{\text{seed}}$. Steps 3.1-3.3 apply only sound implications guarded by $\Sigma$.

    1. Step 3.1 instantiates Meek's R2-type (Spirtes et al., 2000, Section 2.1.2) acyclicity propagation.

    2. Steps 3.2-3.3 decide collider/non-collider at unshielded triples using the exclusivity of $Z$ across minimal separators (either $Z \in S$ for all $S \in \Sigma(X, Y)$ or $Z \notin S$ for all such $S$). By Lemma I.1, under a perfect oracle, this step yields the complete and correct set of compelled $v$-structures and forbidden ones for unshielded triples (Spirtes et al., 2000, Lemma 5.1.3).

Consider the operator that applies one CI-guarded implication of Step 3 at a time. This operator is monotone (it only adds arrowheads) and preserves consistency; hence termination follows by finiteness. To establish uniqueness of the limit, observe that the CI guards restrict rule applications to those that are sound under $\Sigma$, but do not introduce any new rule beyond Meek's rules. Consequently, the set of reachable PDAGs by exhausting Steps 3.1-3.3 from $\mathcal{G}_0$ coincides with the set of Meek's rule completions of $(\mathcal{G}_0, \mathcal{C}(\Sigma))$. By the confluence and maximality of Meek's rules (Meek, 2013, Theorems 2-3), this completion is unique and maximally oriented given the skeleton and the fixed collider set; therefore every fair application order of Steps 3.1-3.3 converges to the same CPDAG. □

**Proof of Theorem 5.1**    With a perfect CI oracle under Markov and Adjacency-Faithfulness, the skeleton stage returns the true skeleton and records minimal separators in $\Sigma$ (Spirtes et al., 2000, Section 5.1). By Lemma I.1, for each unshielded triple, $\Sigma$ fixes whether the center is (non-)collider. By Lemma I.2, implementing Steps 3 from the seeded PDAG terminates and yields a unique maximally oriented PDAG compatible with the skeleton and $\Sigma \cup E_{\text{seed}}$, independent of rule order. This PDAG therefore has exactly the skeleton and $v$-structures of $G$, hence equals the CPDAG (essential graph) of $G$ (Andersson et al., 1997, Theorem 4.1). Consequently, any remaining undirected edges are reversible, so Step 4 performs no orientations.

If $E_{\text{seed}} = \emptyset$, Step 3 applies the same set of rules on the same skeleton and $\Sigma$ as PC/PC-stable/CPC; by confluence of Meek's rules, the closure matches theirs.

## J    PROOFS FOR 5.2

Let $S_{Z,\ell}$ and $U_{Z,\ell}$ denote the numbers of true sepsets and true non-sepsets in incZ$_\ell$, and let $S_{\neg Z,\ell}$ and $U_{\neg Z,\ell}$ denote the same counts for notZ$_\ell$, and $S_\ell = S_{Z,\ell} + S_{\neg Z,\ell}$, $U_\ell = U_{Z,\ell} + U_{\neg Z,\ell}$.

**Lemma J.1** (First-hit filtration across levels). *Define event $D$ "there is at least one independence hit at some level". Define $F_\ell$ as the event "no hit on levels $k < \ell$, and at least one hit on level $\ell$". Under 5.3 (independent CI tests, FPR $= \alpha$, FNR $= \beta$), we have*

$$\Pr(D) = 1 - \prod_\ell \beta^{S_\ell}(1 - \alpha)^{U_\ell}$$

$$\mathrm{PrevNoHit}_\ell := \Pr(\textit{no hit on } k < \ell) = \prod_{k < \ell} \beta^{S_k}(1 - \alpha)^{U_k}$$

$$\Pr(F_\ell) = \mathrm{PrevNoHit}_\ell \cdot \left(1 - \beta^{S_\ell}(1 - \alpha)^{U_\ell}\right)$$

*where $\{F_\ell\}$ is a disjoint partition of $D$.*

*Proof.* Under Assumption 5.3 each candidate on level $k$ hits (declares independence) with probability $1 - \beta$ if it is a true sepset and with probability $\alpha$ if not; candidates and levels are independent. Thus "no hit anywhere" has probability $\prod_\ell \beta^{S_\ell}(1 - \alpha)^{U_\ell}$, giving $\Pr(D)$. The rest is by independence across levels and the definition of $F_\ell$. $\square$

**Assumption J.2** (No short detours; $Z$-only control up to order $\ell$). *Fix integers $\ell \geq 0$ and $L \geq 2\ell+1$. (a) (No short detours) Every $X \rightsquigarrow Y$ path of length $\leq L$ contains $Z$. (b) ($Z$-only control) For any conditioning set $C$ with $|C| \leq \ell$, the segment $X - Z - Y$ is open/blocked iff $Z \notin C/Z \in C$ when $Z$ is a non-collider, and blocked/open iff $Z \notin C/Z \in C$ when $Z$ is a collider; in particular, conditioning on any other node (including descendants on the segment) cannot change the segment's status for $|C| \leq \ell$.*

**Lemma J.3** (Bucket counts and collider/non-collider separation). *Assume Markov and Faithfulness and Assumption J.2 (with $2\ell + 1 \leq L$), at level $\ell = 0$, if $Z$ is a non-collider, $S_{\neg Z,0} = 0$, $U_{\neg Z,0} = 1$; if $Z$ is a collider, $S_{\neg Z,0} = 1$, $U_{\neg Z,0} = 0$. In both cases, incZ$_0 = \varnothing$, hence $S_{Z,0} = U_{Z,0} = 0$.*

*For all $\ell \geq 1$, if non-collider truth, $S_{Z,\ell} = incZ_\ell$, $U_{Z,\ell} = 0$, $S_{\neg Z,\ell} = 0$, $U_{\neg Z,\ell} = notZ_\ell$; if collider truth, $S_{Z,\ell} = 0$, $U_{Z,\ell} = incZ_\ell$, $S_{\neg Z,\ell} = notZ_\ell$, $U_{\neg Z,\ell} = 0$.*

*Proof.* D-separation rules are that, a non-collider blocks a path iff it is in $C$; a collider blocks unless it or a descendant is in $C$.

When $\ell = 0$, for a non-collider chain $X - Z - Y$, the path is open unconditionally, so $\emptyset$ is a non-sepset; for a collider, it is blocked unconditionally, so $\emptyset$ is a sepset.

When $\ell \geq 1$: by Assumption J.2(a), all short $X - Y$ paths pass through $Z$; by Assumption J.2(b), with $|C| \leq \ell$ the local segment's status is controlled only by the inclusion of $Z$. So for a non-collider, if $Z \in C$ (bucket incZ$_\ell$), the local segment is blocked and, since every short path uses $Z$, all paths are blocked, thus sepset; if $Z \notin C$, the local segment is active and cannot be blocked by other vertices with $|C| \leq \ell$, thus non-sepset. For a collider, if $Z \in C$, the local segment is opened and cannot be re-blocked by $|C| \leq \ell$, thus non-sepset; if $Z \notin C$, the local segment remains blocked and no short detour exists, thus sepset. $\square$

**Lemma J.4** (Order-averaged first-hit factor within a level). *Fix a level containing $m$ true sepsets (each hits with prob. $a = 1 - \beta$) and $n$ non-sepsets (each hits with prob. $b = \alpha$). Under a uniformly random within-level permutation (independent of outcomes by Assumption 5.3), the average no-hit-from-predecessors factor for a fixed candidate equals*

$$I_{m,n}(a,b) = \int_0^1 (1 - au)^m (1 - bu)^n \, du = \sum_{i=0}^m \sum_{j=0}^n \binom{m}{i}\binom{n}{j}\frac{(-a)^i(-b)^j}{i + j + 1}$$

*with $I_{m,0} = \frac{1-(1-a)^{m+1}}{a(m+1)}$ and $I_{0,n} = \frac{1-(1-b)^{n+1}}{b(n+1)}$.*

*Proof.* Let $N := m + n + 1$ be the number of candidates on the level including a fixed target $k$. Write $p_j \in \{a, b\}$ for the hit probability of candidate $j \neq k$ and, for a random permutation $\pi$, define $X_\pi = \prod_{j \prec_\pi k}(1 - p_j)$, the product of "no-hit" factors over predecessors of $k$. We average $X_\pi$ over all permutations.

If $r$ candidates precede $k$, then (i) $r$ is uniform on $\{0, \ldots, N-1\}$ with probability $1/N$; (ii) conditional on $r$, the predecessor set $S$ is uniform over the $\binom{N-1}{r}$ subsets of $\{j \neq k\}$ of size $r$. Hence

$$\mathbb{E}_\pi[X_\pi] = \frac{1}{N} \sum_{r=0}^{N-1} \frac{1}{\binom{N-1}{r}} \sum_{\substack{S \subseteq \{j \neq k\} \\ |S| = r}} \prod_{j \in S} (1 - p_j).$$

Insert

$$\frac{1}{N\binom{N-1}{r}} = \frac{r!(N-1-r)!}{N!} = \int_0^1 u^r (1-u)^{N-1-r} \, du$$

and swap sum/integral (justified since the sums are finite), we have

$$\mathbb{E}_\pi[X_\pi] = \int_0^1 \left[ \sum_{r=0}^{N-1} e_r \, u^r (1-u)^{N-1-r} \right] du$$

where $e_r = \sum_{|S|=r} \prod_{j \in S} (1 - p_j)$ are elementary symmetric sums of $\{1 - p_j\}_{j \neq k}$.

The generating function is

$$\sum_{r=0}^{N-1} e_r t^r = \prod_{j \neq k} \big( 1 + (1 - p_j) t \big)$$

With $t = \frac{u}{1-u}$ and factoring $(1-u)^{N-1}$ we obtain

$$\sum_{r=0}^{N-1} e_r \, u^r (1-u)^{N-1-r} = \prod_{j \neq k} (1 - p_j u)$$

so

$$\mathbb{E}_\pi[X_\pi] = \int_0^1 \prod_{j \neq k} (1 - p_j u) \, du$$

There are $m$ terms with $p_j = a$ and $n$ with $p_j = b$, hence

$$I_{m,n}(a, b) = \int_0^1 (1 - au)^m (1 - bu)^n \, du$$

Expanding $(1 - au)^m (1 - bu)^n$ and integrating termwise yields the stated double sum. The special cases follow by taking $n = 0$ or $m = 0$.

For a specific candidate $k$ with hit probability $p_k \in \{a, b\}$,

$$\Pr(k \text{ is first hit}) = p_k \cdot \mathbb{E}_\pi \left[ \prod_{j \prec_\pi k} (1 - p_j) \right] = \begin{cases} a \, I_{S_\ell - 1, U_\ell}(a, b), & k \text{ a true sepset,} \\ b \, I_{S_\ell, \, U_\ell - 1}(a, b), & k \text{ a non-sepset.} \end{cases}$$

$\square$

**Lemma J.5** (Level-$\ell$ identification probabilities). *Condition on the partition $\{F_\ell\}$ from Lemma J.1. Under Assumption 5.3,*

$$\Pr(E \mid D) = \sum_\ell \frac{\Pr(E \cap F_\ell)}{\Pr(D)} = \sum_\ell \frac{\text{PrevNoHit}_\ell}{\Pr(D)} \cdot \Pr(E \text{ via level } \ell \mid \text{ level } \ell \text{ has a hit}).$$

*For CPC (bucket exclusivity within a level) and PC (first hit within a level),*

$$\Pr_{\text{CPC}}(\text{identified as collider} \mid D) = \frac{1}{\Pr(D)} \sum_\ell \text{PrevNoHit}_\ell \underbrace{\beta^{S_{Z,\ell}} (1-\alpha)^{U_{Z,\ell}}}_{\text{no Z-hits at } \ell} \underbrace{\left( 1 - \beta^{S_{\neg Z,\ell}} (1-\alpha)^{U_{\neg Z,\ell}} \right)}_{\text{some non-Z hit}},$$

$$\Pr_{\text{CPC}}(Z \text{ in all saved sepsets} \mid D) = \frac{1}{\Pr(D)} \sum_\ell \text{PrevNoHit}_\ell \beta^{S_{\neg Z,\ell}} (1-\alpha)^{U_{\neg Z,\ell}} \left( 1 - \beta^{S_{Z,\ell}} (1-\alpha)^{U_{Z,\ell}} \right),$$

$$\Pr_{\text{PC}}(\text{identified as collider} \mid D) = \frac{1}{\Pr(D)} \sum_\ell \text{PrevNoHit}_\ell \left[ S_{\neg Z,\ell} (1-\beta) I_{S_\ell - 1, U_\ell}((1-\beta), \alpha) + U_{\neg Z,\ell} \, \alpha \, I_{S_\ell, U_\ell - 1}((1-\beta), \alpha) \right.$$

$$\Pr_{\text{PC}}(Z \text{ in saved sepsets} \mid D) = \frac{1}{\Pr(D)} \sum_\ell \text{PrevNoHit}_\ell \left[ S_{Z,\ell} (1-\beta) I_{S_\ell - 1, U_\ell}((1-\beta), \alpha) + U_{Z,\ell} \, \alpha \, I_{S_\ell, U_\ell - 1}((1-\beta), \alpha) \right].$$

*Proof.* Condition on $F_\ell$ and apply CPC's exclusivity or PC's first-hit rule with the order-averaged factors from Lemma J.4. $\square$

**Proofs for Theorem 5.5** For CPC, under non-collider truth (sepsets in incZ$_\ell$), Lemma J.3 gives $S_{Z,\ell} = m$, $U_{Z,\ell} = 0$, $S_{\neg Z,\ell} = 0$, $U_{\neg Z,\ell} = n$. Plugging into Lemma J.5 yields $\Pr_{\text{CPC}}(\text{collider at } \ell) = \text{PrevNoHit}_\ell\, \beta^m \big[1 - (1-\alpha)^n\big]$; under collider truth (sepsets in notZ$_\ell$), the symmetric expression is $\Pr_{\text{CPC}}(Z \text{ in saved sepset at } \ell) = \text{PrevNoHit}_\ell\, \beta^n \big[1 - (1-\alpha)^m\big]$. Take the ratio to cancel $\text{PrevNoHit}_\ell$. For CPC, with the same bucket counts and Lemma J.4, $\Pr_{\text{PC}}(\text{collider at } \ell) = \text{PrevNoHit}_\ell \cdot n \cdot b \cdot I_{m,\,n-1}(a,b)$, and $\Pr_{\text{PC}}(Z \text{ in saved sepset at } \ell) = \text{PrevNoHit}_\ell \cdot m \cdot b \cdot I_{n,\,m-1}(a,b)$. Divide to cancel $\text{PrevNoHit}_\ell$ and $b$.

For Level-$\ell$ ($\ell \geq 1$), the wrong-orientation odds under Assumptions 5.3,5.4 can be obtained as

$$\mathcal{R}_\ell^{\text{CPC}} = \beta^{m-n} \frac{1 - (1-\alpha)^n}{1 - (1-\alpha)^m}$$

$$\mathcal{R}_\ell^{\text{PC}} = \frac{n}{m} \cdot \frac{I_{m,\,n-1}(1-\beta,\alpha)}{I_{n,\,m-1}(1-\beta,\alpha)}$$

where $I_{p,q}(a,b) = \int_0^1 (1-au)^p (1-bu)^q\, du$, $m = \text{incZ}_\ell$ and $n = \text{notZ}_\ell$.

Therefore, for early levels $\ell \geq 1$ and $\ell \ll \frac{M}{2}$:

$$\text{CPC: } \mathcal{R}_\ell^{\text{CPC}} \approx \beta^{m-n} \frac{n}{m} = \beta^{\binom{M-1}{\ell-1} - \binom{M-1}{\ell}} \cdot \frac{M-\ell}{\ell} \quad (\alpha \text{ small})$$

$$\text{PC: } \mathcal{R}_\ell^{\text{PC}} \approx \frac{n(n+1)}{m(m+1)} \left[1 + \alpha \frac{(m-n)(m+n+1)}{(m+2)(n+2)}\right] \quad (\alpha, \beta \text{ small}),$$

Taking $\alpha, \beta = o\left(\frac{1}{M}\right)$ completes the proof for Theorem 5.5. As $\binom{M-1}{\ell}$ grows with $\ell$ up to $M/2$, $n > m$, $\beta^{m-n} > 1$ as $\beta < 1$, so $\mathcal{R}_\ell^{\text{CPC}} > 1$. At zeroth order $\mathcal{R}_\ell^{\text{PC}} \approx \frac{n(n+1)}{m(m+1)} \approx \left(\frac{M-\ell}{\ell}\right)^2$ when $m, n$ are large, so $\mathcal{R}_\ell^{\text{PC}} > 1$ for $\ell < M/2$.

**Corollary J.6** (small-$\alpha$ approximation for CPC). *For $\ell \geq 1$ and small $\alpha$, as $1 - (1-\alpha)^t = t\alpha + O(\alpha^2)$, we have*

$$\mathcal{R}_\ell^{\text{CPC}} = \beta^{m-n} \frac{1 - (1-\alpha)^n}{1 - (1-\alpha)^m} \approx \beta^{m-n} \frac{n}{m} = \beta^{\binom{M-1}{\ell-1} - \binom{M-1}{\ell}} \cdot \frac{M-\ell}{\ell},$$

*with relative error $O(\alpha)$.*

**Corollary J.7** (small-$\alpha$ and small-$\beta$ approximation for PC). *For $\ell \geq 1$ and small $\alpha, \beta$,*

$$I_{m,n}(1-\beta,\alpha) \approx \frac{1}{m+1} + \frac{\beta}{m+1} - \frac{n\alpha}{(m+1)(m+2)} + O(\alpha^2, \alpha\beta, \beta^2).$$

*Using the ratio expansion $(x + \delta_x)/(y + \delta_y) \approx (x/y)\left[1 + (\delta_x/x) - (\delta_y/y)\right]$, we have*

$$\mathcal{R}_\ell^{\text{PC}} \approx \frac{n(n+1)}{m(m+1)} \left[1 + \alpha \frac{(m-n)(m+n+1)}{(m+2)(n+2)}\right] + O(\alpha^2, \alpha\beta, \beta^2).$$

*Zeroth order (ignore $\alpha, \beta$) is*

$$\mathcal{R}_\ell^{\text{PC}} \approx \frac{n(n+1)}{m(m+1)} = \frac{M-\ell}{\ell} \cdot \frac{n+1}{m+1}.$$

*and $\frac{n+1}{m+1}$ has no tidy closed form in $M, \ell$. For large $m, n$, $\frac{n+1}{m+1} \approx \frac{n}{m}$, giving $\mathcal{R}_\ell^{\text{PC}} \approx \left(\frac{M-\ell}{\ell}\right)^2$.*

*Proof.*

$$I_{m,n}(1-\beta,\alpha) = \int_0^1 \big(1 - (1-\beta)u\big)^m \big(1 - \alpha u\big)^n\, du.$$

Write

$$\left(1 - (1-\beta)u\right)^m = (1-u)^m \left(1 + \frac{\beta u}{1-u}\right)^m \approx (1-u)^m \left(1 + m\frac{\beta u}{1-u}\right),$$

$$(1 - \alpha u)^n \approx 1 - n\alpha u,$$

keeping terms up to $O(\alpha, \beta)$ and dropping $O(\alpha^2, \alpha\beta, \beta^2)$ then multiplying (and ignoring the $\alpha\beta$ cross-term):

$$\left(1 - (1-\beta)u\right)^m (1 - \alpha u)^n \approx (1-u)^m + m\beta\, u(1-u)^{m-1} - n\alpha\, u(1-u)^m.$$

Integrate termwise, we have

$$I_{m,n}(1 - \beta, \alpha) \approx \frac{1}{m+1} + \frac{\beta}{m+1} - \frac{n\,\alpha}{(m+1)(m+2)} + O(\alpha^2, \alpha\beta, \beta^2),$$

Since

$$\mathcal{R}_\ell^{\mathrm{PC}} = \frac{n}{m} \cdot \frac{I_{m,\,n-1}(1-\beta, \alpha)}{I_{n,\,m-1}(1-\beta, \alpha)}, \quad m = \mathrm{incZ}_\ell,\ n = \mathrm{notZ}_\ell.$$

Apply the expansion:

$$I_{m,\,n-1}(1-\beta, \alpha) \approx \frac{1}{m+1} + \frac{\beta}{m+1} - \frac{(n-1)\alpha}{(m+1)(m+2)},$$

$$I_{n,\,m-1}(1-\beta, \alpha) \approx \frac{1}{n+1} + \frac{\beta}{n+1} - \frac{(m-1)\alpha}{(n+1)(n+2)}.$$

Use the first-order ratio expansion $\frac{x+\delta_x}{y+\delta_y} \approx \frac{x}{y}\left[1 + \frac{\delta_x}{x} - \frac{\delta_y}{y}\right]$, where $x = \frac{1}{m+1}$, $y = \frac{1}{n+1}$. As $\frac{\delta_x}{x} = \beta - \frac{(n-1)\alpha}{m+2}$, $\frac{\delta_y}{y} = \beta - \frac{(m-1)\alpha}{n+2}$, the ratio is approximated by $\frac{n+1}{m+1}\left[1 + \alpha\left(\frac{m-1}{n+2} - \frac{n-1}{m+2}\right)\right]$.

Therefore

$$\mathcal{R}_\ell^{\mathrm{PC}} \approx \frac{n(n+1)}{m(m+1)}\left[1 + \alpha\frac{(m-n)(m+n+1)}{(m+2)(n+2)}\right] + O(\alpha^2, \alpha\beta, \beta^2).$$

At zeroth order (ignore $\alpha, \beta$),

$$\mathcal{R}_\ell^{\mathrm{PC}} \approx \frac{n(n+1)}{m(m+1)} = \left(\frac{n}{m}\right)\left(\frac{n+1}{m+1}\right).$$

Since $n/m = \binom{M-1}{\ell}/\binom{M-1}{\ell-1} = \frac{M-\ell}{\ell}$,

$$\mathcal{R}_\ell^{\mathrm{PC}} \approx \frac{M-\ell}{\ell} \cdot \frac{n+1}{m+1},$$

and $\frac{n+1}{m+1}$ has no tidy closed form in $M, \ell$. For large $m, n$, $\frac{n+1}{m+1} \approx \frac{n}{m}$, giving $\mathcal{R}_\ell^{\mathrm{PC}} \approx \left(\frac{M-\ell}{\ell}\right)^2$. $\qquad\square$

# K    NUMERICAL EXPERIMENTS ON NON-COLLIDER / COLLIDER IDENTIFICATION FPRs

We set max searching layer as $l = 3$, $\alpha = 0.05$ and $\beta = 0.1$, plugging in the number of nodes, arcs and average degrees of datasets, we can accordingly compute the expected FPRs as Table 7.

Table 7: Expected false positive rates (FPRs) during identifications.

| network | PC (colliders-first) | PC (nonc-first) | CPC (colliders-first) | CPC (nonc-first) |
|---|---|---|---|---|
| asia | 0.177849 | 0.000926 | 0.071491 | $5. \times 10^{-8}$ |
| alarm | 0.583846 | 0.000159 | 0.128392 | $5. \times 10^{-37}$ |
| cancer | 0.102895 | 0.001906 | 0.059310 | $6. \times 10^{-5}$ |
| child | 0.399529 | 0.000309 | 0.105279 | $5. \times 10^{-20}$ |
| hailfinder | 0.700605 | 0.000103 | 0.138733 | $5. \times 10^{-56}$ |
| hepar2 | 0.755133 | 0.000082 | 0.141944 | $5. \times 10^{-70}$ |
| insurance | 0.488650 | 0.000222 | 0.117261 | $5. \times 10^{-27}$ |
| mildew | 0.567216 | 0.000168 | 0.126597 | $5. \times 10^{-35}$ |
| water | 0.540173 | 0.000185 | 0.123536 | $5. \times 10^{-32}$ |
| win95pts | 0.773360 | 0.000075 | 0.142753 | $5. \times 10^{-76}$ |

## L  PROMPTING TEMPLATES AND PARSING RULE

### L.1  ANSWER TAG PARSING

We extract the final choice using the following case-insensitive regular expression, which returns a single capital letter in {A,B,C,D,E}:

Listing 1: Regex for parsing the <Answer> tag.

```
_ans_re = re.compile(r"<\s*answer\s*>\s*([ABCDE])\s*<\s*/\s*answer\s*>",
    re.I)
```

### L.2  PROMPT TEMPLATES

Placeholders in braces are programmatically substituted (e.g., {u}, {v}, {data_desc}).

Listing 2: Full chain-of-thought selection template.

```
You are a senior researcher in causal discovery. We are studying the
    following dataset:

{data_desc}

The two target variables under review are {u} and {v}.

Conditional-independence tests mentioning these variables:

{ci_bullets}

Neighbour chain(s) that must normally remain non-collider:

{chains}

The nodes involved are described as below:

{node_desc}

Choose one explanation that best fits domain knowledge and/or decides a
    CI test is unreliable (avoid selecting D or E unless other options
    are strongly against common sense):

A. Undecided. We don't know enough to confidently pick a directionality.
B. Changing the state of {u} causally affects {v}, and {v} causally
    affects {u_theOther_2v}.
C. Changing the state of {v} causally affects {u}, and {u} causally
    affects {v_theOther_2u}.
D. Changing the state of {u} causally affects {v}, and {u_theOther_2v}
    also causally affects {v}, **violating corresponding CI tests**.
E. Changing the state of {v} causally affects {u}, and {v_theOther_2u}
    also causally affects {u}, **violating corresponding CI tests**.

Think step-by-step before selecting:
1. Mechanisms - What known causal pathways (biological, physical, etc.)
    support each direction?
2. Counterfactual test - What would happen if we intervened on one node?
    What would we expect?
3. Empirical check - Point to one key piece of information that favors/
    weakens a direction.
4. Comparison - Briefly weigh A vs B vs C vs D vs E and choose the most
    plausible.

Return exactly three lines:
1. Reasoning in support of one direction.
2. Reasoning against the weaker/less plausible direction.
```

```
3. Final choice:  <Answer>A/B/C/D/E</Answer>
```

## L.3 TEMPLATE WHEN ONLY $v \to u$ ANCILLARY EDGE IS POSSIBLE (_CHAIN_PROMPT_TMPL_NONE2U)

Listing 3: Restricted template (None2u).

```
You are a senior researcher in causal discovery. We are studying the
    following dataset:

{data_desc}

The two target variables under review are {u} and {v}.

Conditional-independence tests mentioning these variables:

{ci_bullets}

Neighbour chain(s) that must normally remain non-collider:

{chains}

The nodes involved are described as below:

{node_desc}

Choose one explanation that best fits domain knowledge and/or decides a
    CI test is unreliable (avoid selecting D unless other options are
    strongly against common sense):

A. Undecided. We don't know enough to confidently pick a directionality.
B. Changing the state of {u} causally affects {v}, and {v} causally
    affects {u_theOther_2v}.
C. Changing the state of {v} causally affects {u}.
D. Changing the state of {u} causally affects {v}, and {u_theOther_2v}
    also causally affects {v}, **violating corresponding CI tests**.

Think step-by-step before selecting:
1. Mechanisms - What known causal pathways (biological, physical, etc.)
    support each direction?
2. Counterfactual test - What would happen if we intervened on one node?
    What would we expect?
3. Empirical check - Point to one key piece of information that favors/
    weakens a direction.
4. Comparison - Briefly weigh A vs B vs C vs D and choose the most
    plausible.

Return exactly three lines:
1. Reasoning in support of one direction.
2. Reasoning against the weaker/less plausible direction.
3. Final choice:  <Answer>A/B/C/D</Answer>
```

## L.4 TEMPLATE WHEN ONLY $u \to v$ ANCILLARY EDGE IS POSSIBLE (_CHAIN_PROMPT_TMPL_NONE2V)

Listing 4: Restricted template (None2v).

```
You are a senior researcher in causal discovery. We are studying the
    following dataset:

{data_desc}
```

```
The two target variables under review are {u} and {v}.

Conditional-independence tests mentioning these variables:

{ci_bullets}

Neighbour chain(s) that must normally remain non-collider:

{chains}

The nodes involved are described as below:

{node_desc}

Choose one explanation that best fits domain knowledge and/or decides a
    CI test is unreliable (avoid selecting D unless other options are
    strongly against common sense):

A. Undecided. We don't know enough to confidently pick a directionality.
B. Changing the state of {u} causally affects {v}.
C. Changing the state of {v} causally affects {u}, and {u} causally
    affects {v_theOther_2u}.
D. Changing the state of {v} causally affects {u}, and {v_theOther_2u}
    also causally affects {u}, **violating corresponding CI tests**.

Think step-by-step before selecting:
1. Mechanisms - What known causal pathways (biological, physical, etc.)
    support each direction?
2. Counterfactual test - What would happen if we intervened on one node?
    What would we expect?
3. Empirical check - Point to one key piece of information that favors/
    weakens a direction.
4. Comparison - Briefly weigh A vs B vs C vs D and choose the most
    plausible.

Return exactly three lines:
1. Reasoning in support of one direction.
2. Reasoning against the weaker/less plausible direction.
3. Final choice:  <Answer>A/B/C/D</Answer>
```

## L.5 TEMPLATE WITHOUT CI/NEIGHBOUR CONTEXT (_CHAIN_PROMPT_TMPL_NONE)

Listing 5: Minimal template (None).

```
You are a senior researcher in causal discovery. We are studying the
    following dataset:

{data_desc}

The two target variables under review are {u} and {v}.

The nodes involved are described as below:

{node_desc}

Choose one explanation that best fits domain knowledge:

A. Undecided. We don't know enough to confidently pick a directionality.
B. Changing the state of {u} causally affects {v}.
C. Changing the state of {v} causally affects {u}.

Think step-by-step before selecting:
1. Mechanisms - What known causal pathways (biological, physical, etc.)
    support each direction?
```

```
2. Counterfactual test - What would happen if we intervened on one node?
    What would we expect?
3. Empirical check - Point to one key piece of information that favors/
    weakens a direction.
4. Comparison - Briefly weigh A vs B vs C and choose the most plausible.

Return exactly three lines:
1. Reasoning in support of one direction.
2. Reasoning against the weaker/less plausible direction.
3. Final choice:  <Answer>A/B/C</Answer>
```

## M    REASONING EXAMPLES

Below are 2 examples of the LLM reasoning trace in support of a direction. The question asked is from Appendix L2.

Listing 6: Reasoning with large p-values

```
Reasoning in support of one direction.
B is supported by domain knowledge: 'either' (tuberculosis or lung cancer
    ) can cause dyspnoea as a symptom. Intervening on 'either' (e.g.,
    treating the disease) would alter dyspnoea, aligning with a causal
    direction from 'either' to dyspnoea. The neighbor chain being non-
    collider further supports this directionality.

Reasoning against the weaker/less plausible direction.
C and D contradict domain knowledge: dyspnoea is an effect, not a cause,
    of diseases like tuberculosis or lung cancer. The CI tests (p-values
    ~0.7-1) indicate no direct link between dyspnoea and the other
    variables given 'either', invalidating claims of dyspnoea influencing
     'either' or its dependencies.

Final choice: <Answer>B</Answer>
```

Listing 7: Reasoning with small p-values

```
1. Reasoning in support of one direction.
Stroke Volume (SV) directly determines Cardiac Output (CO) via the
    relationship CO = SV x Heart Rate. CO, in turn, is a primary
    determinant of Blood Pressure (BP), as BP ~= CO x Peripheral
    Resistance. The CI tests confirm that once CO is known, SV and BP
    become independent (removing the Stroke Volume-Blood Pressure edge),
    and Hypovolemia/Left-Ventricular Failure influence CO only via SV (
    removing their direct links to CO). This aligns with the
    physiological mechanism where SV -> CO -> BP.

2. Reasoning against the weaker/less plausible direction.
Option C reverses the causal relationship between CO and SV, which
    contradicts the physiological definition of CO as a product of SV and
     Heart Rate. Additionally, Stroke Volume influencing diagnostic
    variables like Hypovolemia is implausible, as these conditions are
    root causes (diagnostic variables have no predecessors). Options D
    and E imply violations of CI tests, which are explicitly rejected by
    the p-values, leaving B as the only direction consistent with both
    domain knowledge and statistical evidence.

3. Final choice: <Answer>B</Answer>
```

# N  ASSUMPTIONS OF RELATED WORK

**Vashishtha** (Vashishtha et al., 2025) treat the LLM as an imperfect expert queried for causal direc-
tion, and assume causal order (a topological order) is a more stable interface for expert knowledge
("we propose that causal order be used as a more stable output interface"), as opposed to directly
prompting undirected edges for their orientations. Pairwise LLM querying produces inconsistencies
("pairwise prompts lead to multiple cycles"), so they add an explicit local consistency constraint by
prompting on triplets and "instructing the LLM to avoid cycles within this triplet," then vote over
redundant triplets; they also analyze an "imperfect expert with an error $\epsilon$ on each prediction" and
assumes "the $\epsilon$ expert to have error probability exactly equal to $\epsilon$".

**Causal Disco** (Long et al., 2023) explicitly models the LLM as an imperfect expert oracle
($E(\mu_i, \mu_j)$) that consumes metadata (names/descriptions) and returns a hypothesized orientation
for an edge. A core assumption of their setup is that the true Markov equivalence class is already
available ("We assume that ($M_{G^\star}$) is known, e.g., that it has been obtained via some causal dis-
covery algorithm"), and the LLM is used only to orient edges that are ambiguous within that MEC.
They further assume a simple noise model for LLM mistakes: the expert's response depends only
on the true orientation and is "incorrect with constant probability ($\epsilon$)," which enables Bayesian rea-
soning over consistent orientations while controlling risk (they target keeping ($G^\star$) in the remaining
set with probability at least $(1 - \eta)$).

**LLM-CD** (Ban et al., 2025) is another imperfect expert method. Concretely, they assume LLM
reasoning should be restricted ("an accuracy-oriented prompting strategy restricts causal analysis
to a reliable range"), because "LLM-based causal reasoning on all pairs. . . [is]" unreliable when
mechanisms are unclear. Instead of edge constraints they translate LLM outputs into ancestral/path
constraints: "We do not directly specify the existence of edges. . . [LLM] knowledge [is] qualitative
and can represent indirect interactions." A key consideration of this approach is the reliance on a
manually tuned confidence threshold for determining LLM accuracy, which as shown in their paper
may vary between datasets.

**Jiralerspong** (Jiralerspong et al., 2024) assumes an LLM can answer causal queries from metadata
alone and explicitly "utilize LLMs to respond to causal queries solely based on metadata. . . without
accessing the numerical observations," analogizing this to how human experts build graphs from
descriptions. Their main additional assumption for scalability is that full-graph discovery can be
organized as a traversal: they construct the graph via BFS with only (O(n)) prompts by asking (i)
root-node style questions ("Ask which variables are not caused by any other variables") and (ii)
expansion questions ("ask which variables it causes"), inserting edges unless they "form a cycle" to
enforce the DAG constraint.

**ILS-CSL** (Ban et al., 2023) assumes LLM judgments are most reliable when scoped to checking
candidate edges suggested by data. Prior LLM-for-CSL approaches can yield "unreliable constraints
from imperfect LLM inferences" and that full pairwise prompting of all nodes in the graph is costly;
their key design choice is to "focus LLMs on verifying direct causal relationships already suggested
by the data," i.e., use LLMs as an iterative supervisor of a data-driven CSL loop. They assume
convergence can be defined operationally: "The iteration concludes when the LLM-based inferences
and data-driven CSL align".

**SCP** (Takayama et al., 2024) assumes LLMs can contribute background causal knowledge without
needing data, and that this knowledge becomes more useful when conditioned on statistical signals
from an initial causal discovery run. The LLM is "prompted with the results of the SCD" and "eval-
uates the probability of the causal relationships considering both the domain knowledge and the
statistical characteristics suggested by SCD." They assume pairwise causal answers can be opera-
tionalized as a prior: the LLM "judges whether causal relationships exist between all pairs. . . with
'yes' or 'no'," and "probabilities of the responses. . . [are] transformed into the prior knowledge
matrix. . . reapplied to SCD."

