# OpenReview forum: "Improving constraint-based discovery with robust propagation and reliable LLM priors"
_ICLR.cc/2026/Conference — Submitted to ICLR 2026_

### Official Review · Reviewer_Mmpx · 2025-10-18

**Soundness:** 2
**Presentation:** 2
**Contribution:** 2
**Rating:** 2
**Confidence:** 4

**Summary:**

This paper proposes a causal discovery algorithm called MosaCD. MosaCD improves constraint based causal discovery methods using LLMs and non-collider-identification first approach. Techniques such as shuffling queries are employed to address hallucination problems in LLMs. Experiments are performed on several datasets to show the advantages of MosaCD compared to baselines.

**Strengths:**

1. The paper tackles an important problem of causal discovery
2. The paper is mostly written well and easy to understand

**Weaknesses:**

Major:

1. In the abstract and related work, the paper states that existing methods using LLMs for causal discovery assume perfect experts. However, methods that treat LLMs as imperfect/noisy experts do exist (see [1–3]).

2. As stated in Section 3, MosaCD relies on the idea that “estimated conditional independences can indeed be due to limited power of statistical tests.” However, the MosaCD algorithm starts from a skeleton obtained using conditional-independence tests (beginning with a complete graph). How is reliance on the skeleton obtained from CI tests justified?

3. In Line 5 of the algorithm, when there exists a semi-directed path between $X$ and $Y$ and also  an undirected edge $X-Y$, what is the intuition behind orienting $X\to Y$

4. In the MosaCD algorithm, LLMs are provided with p-values and conditional-independence statements. How can an LLM effectively make use of p-values?

4. In the experiments, MosaCD is not compared against other families of causal-discovery methods such as score-based and continuous optimization–based methods. Also, some LLM-based methods (e.g., [1,2]) are not included as baselines.

5. Why is the structural Hamming distance (SHD) metric not reported in the results? Why is only one base LLM (GPT-4o-mini) used? Why is “Meek” considered a separate baseline—shouldn’t it be part of the PC algorithm?


Minor:

1. Explicitly state the assumptions about the underlying causal graph, such as the no-hidden-confounding assumption.

2. In the introduction, it is written that constraint-based methods are computationally efficient. However, constraint-based methods can be computationally inefficient: when the graph size is large, the number of conditional-independence tests required for graph recovery can be prohibitively large. Also, it is written that constraint-based methods are state-of-the-art. Due to computational challenges, alternatives such as score-based methods and optimization-based methods are often used.


References:

[1] Long, Stephanie, et al. "Causal discovery with language models as imperfect experts."

[2] Vashishtha, Aniket, et al. "Causal inference using LLM-guided discovery."

[3] Ban, Taiyu, et al. "From query tools to causal architects: Harnessing large language models for advanced causal discovery from data."

**Questions:**

See the weaknesses section and provide the responses.

---

> ### Author Response · Authors · 2025-11-24
> **Response to Reviewer Mmpx (1/5)**
>
> Thank you for recognizing the importance of the problem addressed in our work. We appreciate the reviewer’s detailed feedback and have revised the manuscript extensively in response. To address the concerns raised, we have substantially clarified the exposition, added missing theoretical justifications, incorporated 8 additional baseline methods spanning score-based, continuous-optimization, and recent LLM-integrated approaches, and expanded the evaluation metrics (including SHD and multiple open-source LLMs). These revisions include 2 new analyses and updates to 2 main figures/tables, and 2 supplementary figures/tables. We address each comment point by point below.
>
> The reviewer has raised 8 comments/weaknesses/questions. We believe that we have completely addressed 7 out of these 8 points (W1,W2,W3,W5,W6,W7,W8) and substantially addressed the remaining points (W4). Please let us know if there are any additional questions or if further clarification would be helpful. If the reviewer feels that their concerns have been adequately addressed, we would be grateful if they would consider updating their evaluation.
>
> Please see our updated main results table (Table 1) below with the 8 additional baselines. (F1 scores across datasets).
>
> | Method             | Cancer (5) | Asia (8) | Child (20)  | Insurance (27) | Water (32)  |
> | ------------------ | ---------- | -------- | ----------- | -------------- | ----------- |
> | **PC**             | 0.50       | 0.67     | 0.70        | 0.62           | 0.45        |
> | **Meek**           | 0.50       | 0.67     | 0.78        | 0.70    | 0.57        |
> | **Shapley-PC**     | **1.00**   | 0.53     | 0.67        | 0.67           | 0.47        |
> | **ILS-CSL***       | 0.50       | **0.93** | 0.83| 0.70    |0.60 |
> | **SCP***           | 0.50       | 0.67     | 0.78        | 0.68           | 0.57        |
> | **Jiralerspong***  | 0.57       | 0.55     | 0.10        | 0.21           | 0.00        |
> | **Vashishtha***    | 0.75       | 0.93     | 0.67        | 0.70    | 0.65        |
> | **Causal Disco***  | 0.57       | 0.86     | 0.56        | 0.34           | **0.87**    |
> | **LLM-CD***        | **1.00**   | 0.80     | 0.40        | 0.29           | 0.24        |
> | **NOTEARS**        | 0.00       | 0.57     | 0.48        | 0.15           | 0.29        |
> | **DAG-GNN**        | 0.00       | 0.27     | 0.24        | 0.20           | 0.26        |
> | **GES**            | 0.50       | 0.12     | 0.17        | 0.26           | 0.29        |
> | **HC**             | 0.50       | 0.12     | 0.33        | 0.33           | 0.36        |
> | **MosaCD*** (Ours) | **1.00**   | **0.93** | **0.90**    | **0.87**       | 0.59        |
>
> | Method             | Mildew (35) | Alarm (37)  | Hailfinder (56) | Hepar2 (70) | Win95pts (76) |
> | ------------------ | ----------- | ----------- | --------------- | ----------- | ------------- |
> | **PC**             | 0.63        | 0.85        | 0.38            | 0.36        | 0.59          |
> | **Meek**           | 0.69        | 0.90 | 0.40            | 0.39        | 0.64          |
> | **Shapley-PC**     | 0.75        | 0.84        | 0.38            | 0.44        | 0.65          |
> | **ILS-CSL***       | 0.89 | 0.85        | 0.44            | 0.54        | 0.69          |
> | **SCP***           | 0.69        | 0.87        | 0.39            | 0.38        | 0.63          |
> | **Jiralerspong***  | 0.07        | 0.09        | 0.10            | 0.16        | 0.03          |
> | **Vashishtha***    | 0.78        | 0.72        | **0.60**        | 0.66        | 0.74   |
> | **Causal Disco***  | 0.61        | 0.51        | 0.33            | **0.81**    | 0.55          |
> | **LLM-CD***        | 0.24        | 0.30        | 0.22            | 0.32        | 0.17          |
> | **NOTEARS**        | 0.25        | 0.25        | 0.27            | 0.06        | 0.14          |
> | **DAG-GNN**        | 0.43        | 0.53        | 0.13            | 0.13        | 0.37          |
> | **GES**            | 0.23        | 0.26        | 0.33            | 0.45        | 0.51          |
> | **HC**             | 0.24        | 0.30        | 0.31            | 0.47        | 0.46          |
> | **MosaCD*** (Ours) | **0.90**    | **0.93**    | 0.49    | 0.72 | **0.81**      |
> *F1 scores across datasets and methods*

---

> ### Author Response · Authors · 2025-11-24
> **Response to Reviewer Mmpx (2/5)**
>
> > In the abstract and related work, the paper states that existing methods using LLMs for causal discovery assume perfect experts. However, methods that treat LLMs as imperfect/noisy experts do exist (see [1–3]).
>
> Mmpx-W1: related LLM methods.
>
> Thank you for raising this point. The reviewer is correct that several recent works treat LLMs as imperfect/noisy experts. However, these approaches typically rely on strong or unrealistic assumptions about LLM accuracy. Long et al. [1] assumes “an expert that returns correct orientations with some fixed probability”; Vashistha et al. [2] assumes “the ϵ-expert to have error probability exactly equal to ϵ” and given a question with 3 options that “the probability for the two wrong options are equally likely”; and Ban et al. [3] relies on manually setting a “confidence threshold” which can vary greatly in prior accuracy at different thresholds, as reported in their paper. These assumptions are often difficult to satisfy in practice and desired confidence levels will vary, especially given the well-documented instability of LLMs in causal reasoning and positional bias in question answering.
>
> In contrast, our framework does not assume any explicit accuracy level, requires no confidence tuning, and is designed to remain robust under substantial LLM noise. Empirically, we conducted new experiments and found that MosaCD substantially outperformed the 3 imperfect expert LLM methods (top performance on 7/10 datasets and best performance overall, average F1 score 0.81 versus next best Vashishtha 0.69). We also show that MosaCD remains stable even under uninformative variable descriptions that induce large LLM errors (Figure 2 of the original submission).
>
> We have updated the Abstract and Section 1.1 to phrase this better, “implicitly assume perfect experts” → “assume perfect experts or predictable error rates” (lines 18-19, 120-122).
>
> [1] Long, Stephanie, et al. "Causal discovery with language models as imperfect experts."
>
> [2] Vashishtha, Aniket, et al. "Causal inference using LLM-guided discovery."
>
> [3] Ban, Taiyu, et al. "From query tools to causal architects: Harnessing large language models for advanced causal discovery from data."
>
> >As stated in Section 3, MosaCD relies on the idea that “estimated conditional independences can indeed be due to limited power of statistical tests.” However, the MosaCD algorithm starts from a skeleton obtained using conditional-independence tests (beginning with a complete graph). How is reliance on the skeleton obtained from CI tests justified?
>
> Mmpx-W2: reliability of skeleton.
>
> Thank you for the comment. The reviewer is correct that both skeleton construction and collider orientation rely on CI tests, which can be imperfect under limited power. However, skeleton estimation is substantially more stable than collider orientation. Our point in Section 3 is not that CI tests are unusable in general, but that their fine-grained use for identifying separating sets (related to collider orientation) is much more fragile than their coarse use for adjacency discovery (related to skeleton construction).
>
> CI tests—even when noisy—tend to provide reasonably reliable information for adjacency discovery. Skeleton construction only asks whether there exists a conditioning set that renders X ⟂ Y;  occasional CI errors typically add or remove a small number of edges but do not produce systematic distortions of the skeleton. In contrast, collider orientation depends on whether a specific variable Z is included or excluded from separating sets, a decision that is often highly unstable when CI tests have limited power.
>
> MosaCD is designed precisely for this regime: we take a skeleton that is “reasonably good” (e.g., from PC, PC-stable, or CPC) and replace the fragile initial collider-orientation step with more robust LLM-based seeds (Step 2), orienting as much of the graph as possible before falling back on collider-based rules (Step 3.3). As shown in our benchmarks, this substantially reduces orientation errors relative to starting collider orientation directly from CI-derived separating sets, while still leveraging the generally reliable adjacency information provided by CI-based skeletons. MosaCD uses collider orientation substantially less than PC variants (avg 3.3 versus PC 28.5). We have updated the manuscript Section 3 (lines 160–161) to clarify this distinction and to justify our reliance on a CI-derived skeleton.

---

> ### Author Response · Authors · 2025-11-24
> **Response to Reviewer Mmpx (3/5)**
>
> >In Line 5 of the algorithm, when there exists a semi-directed path between  and  and also an undirected edge , what is the intuition behind orienting
>
> Mmpx-W3: Intuition behind orienting in line 5.
>
> Thank you for the comment. This step follows Meek’s Rule 2 from Meek (1995). The intuition is that if there already exists a semi-directed path from X to Y (i.e., a path that can only be oriented X $\rightsquigarrow$ Y any DAG completion), then orienting the undirected edge X—Y as Y→X would introduce a directed cycle X $\rightsquigarrow$ Y → X in some completion of the graph. Because a DAG cannot contain such a cycle, all DAGs in the Markov equivalence class must orient this edge as X→Y. This is exactly the scenario illustrated in Figure 1 of Meek (1995). We have updated Section 4 (lines 214-215) to clarify this.
>
> >In the MosaCD algorithm, LLMs are provided with p-values and conditional-independence statements. How can an LLM effectively make use of p-values?
>
> Mmpx-W4: How does LLMs use the p-values and conditional-independence statements.
>
> Thank you for the question. In our setting, p-values provide the LLM with quantitative cues that help guide its judgment in cases where domain knowledge alone is ambiguous. In practice, we observe two main uses: (i) when the LLM is uncertain, p-values help reinforce or weaken candidate causal directions; and (ii) highly significant p-values can override an initially incorrect LLM intuition. We provide 2 examples below:
>
> (i) Example when the LLM is uncertain, p-values help reinforce or weaken candidate causal directions
>
> “C and D contradict domain knowledge: dyspnoea is an effect, not a cause, of diseases like tuberculosis or lung cancer. The CI tests (p-values ~0.7–1) indicate no direct link between dyspnoea and the other variables given 'either', invalidating claims of dyspnoea influencing 'either' or its dependencies.”
>
> (ii) Example where highly significant p-values can override an initially incorrect LLM intuition
>
> “ Option C reverses the causal relationship between CO and SV, which contradicts the physiological definition of CO as a product of SV and Heart Rate. Additionally, Stroke Volume influencing diagnostic variables like Hypovolemia is implausible, as these conditions are root causes (diagnostic variables have no predecessors). Options D and E imply violations of CI tests, which are explicitly rejected by the p-values, leaving B as the only direction consistent with both domain knowledge and statistical evidence.“
>
> We have updated Appendix M to add this example and clarify these points.
>
> >In the experiments, MosaCD is not compared against other families of causal-discovery methods such as score-based and continuous optimization–based methods. Also, some LLM-based methods (e.g., [1,2]) are not included as baselines.
>
> Mmpx-W5: baselines.
>
> Thank you for the comment. We agree that a comprehensive comparison is important. We have added 8 new baselines to our evaluation, including5 requested by the reviewer (NOTEARS, DAG-GNN, Vashistha et al.[2], Causal Disco[1], LLM-CD[4]), and additional LLM method Jiralerspong et al. [3], and 2 additional score-based methods (GES and HC). Results for these expanded baselines are reported in Table 1 at the beginning. MosaCD achieves top performance on 7/10 tasks relative to these 8 new baselines and the best performance overall (average MosaCD F1 score 0.81 versus next best 0.69), demonstrating its superior performance. We have updated Section 6.1 (lines 370-380) to report these findings.
>
> [1] Long, Stephanie, et al. "Causal discovery with language models as imperfect experts."
>
> [2] Causal Order: The Key to Leveraging Imperfect Experts in Causal Inference (Vashishtha et al.,2025)
>
> [3] Efficient Causal Graph Discovery Using Large Language Models (Jiralerspong et al., 2024)
>
> [4] Integrating Large Language Model for Improved Causal Discovery (Ban et al., 2025) (Note this is the same as the [3] you mentioned in your review)

---

> ### Author Response · Authors · 2025-11-24
> **Response to Reviewer Mmpx (4/5)**
>
> >Why is the structural Hamming distance (SHD) metric not reported in the results? Why is only one base LLM (GPT-4o-mini) used? Why is “Meek” considered a separate baseline—shouldn’t it be part of the PC algorithm?
>
> Mmpx-W6: Clarifications.
>
> The reviewer raises 3 questions, which we address in turn.
>
> (i) Structural Hamming distance (SHD)
>
> Thank you for this comment. We agree that including additional metrics provides a more comprehensive assessment. In response, we have added Structural Hamming Distance (SHD) to Appendix B. SHD leads to similar qualitative conclusions as the F1 metric: MosaCD continues to outperform the existing baselines in the same 7/10 datasets. We have updated the manuscript accordingly (lines 375-376, Appendix B).
>
> | Method             | Cancer (5) | Asia (8) | Child (20) | Insurance (27) | Water (32) |
> | ------------------ | ---------- | -------- | ---------- | -------------- | ---------- |
> | **PC**             | 2.0        | 3.0      | 10.5       | 28.0           | 53.0       |
> | **Meek**           | 2.0        | 3.0      | 6.0        | 19.0           | 38.0       |
> | **Shapley-PC**     | **0.0**    | 4.0      | 19.0       | 27.5           | 52.5       |
> | **ILS-CSL***       | 2.0        | **1.0**  | 5.5        | 25.0           | 37.5       |
> | **SCP***           | 2.0        | 3.0      | 6.0        | 21.0           | 38.0       |
> | **Jiralerspong***  | 3.0        | 5.0      | 38.0       | 52.0           | 71.0       |
> | **Vashishtha***    | 1.0        | **1.0**  | 9.0        | 21.0           | 34.0       |
> | **Causal Disco***  | 2.0        | 2.0      | 13.0       | 41.0           | **15.0**   |
> | **LLM-CD***        | **0.0**        | 2.0      | 30.0       | 59.0           | 81.0       |
> | **DAG-GNN**        | 4.0        | 9.0      | 23.0       | 60.0           | 55.0       |
> | **GES**            | 2.0        | 9.0      | 63.0       | 105.0          | 71.0       |
> | **HC**             | 2.0        | 9.0      | 51.0       | 95.0           | 60.0       |
> | **MosaCD*** (Ours) | **0.0**    | **1.0**  | **2.0**    | **12.0**       | 36.0       |
>
> | Method             | Mildew (35) | Alarm (37) | Hailfinder (56) | Hepar2 (70) | Win95pts (76) |
> | ------------------ | ----------- | ---------- | --------------- | ----------- | ------------- |
> | **PC**             | 23.0        | 12.0       | 105.0           | 99.5        | 68.0          |
> | **Meek**           | 17.0        | 7.5        | 97.0            | 87.0        | 57.5          |
> | **Shapley-PC**     | 18.5        | 14.0       | 105.0           | 89.0        | 69.5          |
> | **ILS-CSL***       | 10.0        | 12.0       | 96.5            | 77.5        | 54.0          |
> | **SCP***           | 17.0        | 8.0        | 99.0            | 88.0        | 56.0          |
> | **Jiralerspong***  | 50.0        | 80.0       | 72.0            | 137.0       | 135.0         |
> | **Vashishtha***    | 13.0        | 15.0       | 77.0            | 59.0        | 46.0          |
> | **Causal Disco***  | 24.0        | 27.0       | **53.0**        | **36.0**    | 69.0          |
> | **LLM-CD***        | 52.0        | 59.0       | 92.0            | 117.0       | 160.0         |
> | **DAG-GNN**        | 50.0        | 35.0       | 77.0            | 119.0       | 93.0          |
> | **GES**            | 152.0       | 101.0      | 150.0           | 122.0       | 122.0         |
> | **HC**             | 140.0       | 92.0       | 162.0           | 119.0       | 133.0         |
> | **MosaCD*** (Ours) | **8.0**     | **6.0**    | 90.0            | 52.0        | **37.0**      |
> *SHD scores across datasets and methods*
>
> (ii) LLMs included.
> Thank you for the question. As shown in Section 6.2 of the original submission (Figure 4), we did evaluate MosaCD using 5 different LLMs. In response to the reviewer’s request for broader coverage, we have additionally incorporated results using Qwen-3 32B and Llama-3 8B-Instruct, bringing the total to 7 LLMs. The updated results are reported below and included in the revised manuscript Figure 4 (lines 477-482).
>
>
> | Model                     | Asia | Insurance | Hepar2 |
> | ------------------------- | ---- | --------- | ------ |
> | **Claude 3.5 Haiku**      | 0.93 | 0.68      | 0.67   |
> | **Claude 4 Sonnet**       | 0.93 | 0.87      | 0.70   |
> | **GPT-4o-mini (Default)** | 0.93 | 0.87      | 0.72   |
> | **GPT-5**                 | 0.93 | 0.85      | 0.72   |
> | **Llama-3-8B**            | 0.93 | 0.56      | 0.59   |
> | **GPT-OSS-120B**          | 0.67 | 0.72      | 0.68   |
> | **Qwen3-32B**             | 0.93 | 0.87      | 0.71   |
> *MosaCD F1 scores across LLMs and datasets*

---

> ### Author Response · Authors · 2025-11-24
> **Response to Reviewer Mmpx (5/5)**
>
> Mmpx-W6 continued:
>
> (iii) The “Meek” method.
> Thank you for the question. The “Meek” baseline in our experiments refers to the stand-alone algorithm proposed by Meek (2013) [1]. This is distinct from the Meek orientation rules used within the PC algorithm, which originate from earlier work by Spirtes & Glymour (1991) [2]. Thus, “Meek” in our tables denotes the 2013 causal discovery approach, not the internal Meek rules that PC applies during edge orientation. We have updated Section 6 (lines 352-353) to clarify this.
>
> [1] Meek, Christopher. "Causal inference and causal explanation with background knowledge." arXiv preprint arXiv:1302.4972 (2013).
>
> [2] Spirtes, Peter, and Clark Glymour. "An algorithm for fast recovery of sparse causal graphs." Social science computer review 9.1 (1991): 62-72.
>
> > Explicitly state the assumptions about the underlying causal graph, such as the no-hidden-confounding assumption.
>
> Mmpx-W7: Assumptions about the underlying causal graph.
>
> Thank you for this comment. We follow the standard assumptions used in the PC literature. Specifically, we assume that the true causal structure over the observed variables V is a directed acyclic graph G* with no hidden confounders or selection bias (causal sufficiency). The observational distribution assumed to satisfy the Causal Markov Condition and (Adjacency-)Faithfulness with respect to G*. We have updated Section 5 (lines 249-250) to explicitly state these assumptions.
>
> >In the introduction, it is written that constraint-based methods are computationally efficient. However, constraint-based methods can be computationally inefficient: when the graph size is large, the number of conditional-independence tests required for graph recovery can be prohibitively large. Also, it is written that constraint-based methods are state-of-the-art. Due to computational challenges, alternatives such as score-based methods and optimization-based methods are often used.
>
> Mmpx-W8: Computational efficiency of constraint-based methods.
>
> Thank you for raising this point. The reviewer is correct that constraint-based methods can become computationally expensive on large graphs due to the number of CI tests required, and that scalable score-based or optimization-based alternatives have been developed.
>
> Our intention was not to claim that constraint-based methods are the most computationally efficient, but rather to highlight their distinct advantages (e.g., clear statistical interpretability and principled guarantees under standard assumptions). To avoid confusion, we made two revisions: (i) We removed the phrase “computationally efficient” in the Introduction (lines 38-39), and (ii) we reworded the statement that “constraint-based methods remain state of the art” (line 45-47) to “nevertheless, constraint-based methods remain reliant on v-structure orientation, as no better alternative exists for determining an initial seed set of oriented edges.” to avoid overclaiming and to more accurately reflect we are studying the widely used foundation of initial edge-orientation steps.

---

> > ### Comment · Reviewer_Mmpx · 2025-11-25
> > **Thank you for your response**
> >
> > I thank the authors for the response. I have some followup questions.
> >
> > 1. Regarding W2, how is "skeleton estimation is substantially more stable than collider orientation" justified? The current response is not very convincing. Both Skelton estimation and collider orientation require lot of CI tests that may have limited power. How one is "reasonably good" than the other? Also, the phrase "reasonably good" is not clearly defined.
> >
> > 2. Regarding W3, what is the meaning of "in some completion of the graph"? Consider the example of a semi directed path between $X$ and $Y$: $X - Z - W \to Y$. If we also have $X - Y$, step 3.1 orients $X \to Y$. However, orienting it as $Y\to X$ does not create cycle in one of the completions (e.g., $X \leftarrow Z \leftarrow W \to Y$). What is the justification of missing such possible structure?
> >
> > 3. Regarding W4, Appendix section M is hard to understand. What is the question asked to LLM and what are the options? What are options B/C/D? Please clarify.
> >
> > 4. Regarding W1, if the LLM is imperfect, how does it help in the first place? Without assuming certain level of correctness from LLMs, how do we guarantee the correctness of the final results that depend on the LLMs outputs.

---

> > > ### Author Response · Authors · 2025-11-27
> > > **Response to followup questions (1/2)**
> > >
> > > We thank the reviewer for carefully reading our response. The reviewer has raised 4 additional questions. We address each review question in detail below.
> > >
> > > > Regarding W2, how is "skeleton estimation is substantially more stable than collider orientation" justified? The current response is not very convincing. Both Skelton estimation and collider orientation require lot of CI tests that may have limited power. How one is "reasonably good" than the other? Also, the phrase "reasonably good" is not clearly defined.
> > >
> > > Mmpx-R1-Q1: Justification for skeleton estimation being more stable than collider orientation.
> > >
> > > The reviewer has raised two related questions: (i) further justification for this statement, and (ii) clarification of the phrase “reasonably good.” We address each in turn.
> > >
> > > (i) Further justification for skeleton estimation being more stable than collider orientation.
> > >
> > > First, we provide a concise theoretical explanation using the PC algorithm as an example (see Section 2 of Ramsey 2012, “Adjacency-Faithfulness and Conservative Causal Inference”): In PC, (i) (skeleton estimation) the skeleton is constructed by removing an edge $X-Y$ whenever any separating set $S$ renders $X \perp Y \mid S$, and the corresponding minimal separating set $S$ is recorded in $\Sigma(X,Y)$; and (ii) (collider orientation) for an unshielded triple $X-Z-Y$, the algorithm orients $X \to Z \leftarrow Y$ if $Z \notin \Sigma(X,Y)$, more precisely: if $Z$ is absent from the recorded sepsets $\Sigma(X,Y)$; this is a universal-negative condition.
> > >
> > > There are two reasons why skeleton estimation is more reliable under finite samples, especially when multiple sepsets exist and when the recorded sepsets are incomplete/order-dependent (PC/PC-stable typically store only one, whereas CPC stores multiple). First, skeleton estimation relies on a sufficient condition whereas collider orientation does not: (i) finding any sepset is enough to remove an edge $X-Y$; but (ii) $Z \notin \text{Sepset}(X,Y)$is not sufficient to conclude that for Z being a collider - Z may belong to another valid sepset that the algorithm did not record. Second, skeleton estimation is effectively an “existential” query (“does there exist **at least one** $S$ such that $X \perp Y \mid S$”, needing just one accurate test), whereas collider orientation requires a much more delicate “set-membership” conclusion about sepsets (e.g., proving a node appears in **all / none** of the relevant sepsets, needing all tests to be accurate). Thus, skeleton estimation is inherently more robust than collider orientation.
> > >
> > > Second, we further evaluated both components empirically. Across all datasets in the manuscript, skeleton accuracy is consistently higher (average undirected edge accuracy 92.1% for PC variants) than collider orientation accuracy (average seed accuracy 73.3% for PC variants)
> > >
> > > We recognize that it is our responsibility to provide a clear exposition. We’ve updated Section 3 (lines 161-163) in an effort to improve clarity.
> > >
> > > (ii) clarify the definition of “reasonably good”.
> > >
> > > By “reasonably good,” we simply meant that skeleton estimation is more reliable than collider orientation.
> > >
> > > > Regarding W3, what is the meaning of "in some completion of the graph"? Consider the example of a semi directed path between $X$ and $Y$: $X - Z - W \to Y$. If we also have $X - Y$, step 3.1 orients $X \to Y$. However, orienting it as $Y\to X$ does not create cycle in one of the completions (e.g., $X \leftarrow Z \leftarrow W \to Y$). What is the justification of missing such possible structure?
> > >
> > > Mmpx-R1-Q2: Clarification for W3.
> > >
> > > Thank you for catching this ambiguity. We now realize that this is a typo and should be corrected from “(semi-)directed path” to “directed path”, consistent with the standard Meek R2 rule. Specifically, if there is already a directed path from X to Y (e.g., $X \to Z \to Y$), orienting $X \leftarrow Y$ will create a cycle, and thus $X \to Y$ is the only valid orientation. We have updated line 5 of Algorithm 1 to correct this typo.

---

> > > > ### Author Response · Authors · 2025-11-27
> > > > **Response to followup questions (2/2)**
> > > >
> > > > > Regarding W4, Appendix section M is hard to understand. What is the question asked to LLM and what are the options? What are options B/C/D? Please clarify.
> > > >
> > > > Mmpx-R1-Q3: W4.
> > > >
> > > > Thank you for the comment. In Appendix M, the question asked to LLM is below (same as in Appendix L2).
> > > >
> > > > """
> > > > You are a senior researcher in causal discovery. We are studying the following dataset:
> > > >
> > > > {data_desc}
> > > >
> > > > The two target variables under review are {u} and {v}.
> > > >
> > > > Conditional-independence tests mentioning these variables:
> > > >
> > > > {ci_bullets}
> > > >
> > > > Neighbour chain(s) that must normally remain non-collider:
> > > >
> > > > {chains}
> > > >
> > > > The nodes involved are described as below:
> > > >
> > > > {node_desc}
> > > >
> > > > Choose one explanation that best fits domain knowledge and/or decides a CI test is unreliable (avoid selecting D or E unless other options are strongly against common sense):
> > > >
> > > > A. Undecided. We don't know enough to confidently pick a directionality.
> > > > B. Changing the state of {u} causally affects {v}, and {v} causally affects {u_theOther_2v}.
> > > > C. Changing the state of {v} causally affects {u}, and {u} causally affects {v_theOther_2u}.
> > > > D. Changing the state of {u} causally affects {v}, and {u_theOther_2v} also causally affects {v}, violating corresponding CI tests.
> > > > E. Changing the state of {v} causally affects {u}, and {v_theOther_2u} also causally affects {u}, violating corresponding CI tests.
> > > >
> > > > Think step-by-step before selecting:
> > > > 1. Mechanisms - What known causal pathways (biological, physical, etc.) support each direction?
> > > > 2. Counterfactual test - What would happen if we intervened on one node? What would we expect?
> > > > 3. Empirical check - Point to one key piece of information that favors/weakens a direction.
> > > > 4. Comparison - Briefly weigh A vs B vs C vs D vs E and choose the most plausible.
> > > >
> > > > Return exactly three lines:
> > > > 1. Reasoning in support of one direction.
> > > > 2. Reasoning against the weaker/less plausible direction.
> > > > 3. Final choice:  <Answer>A/B/C/D/E</Answer>
> > > > """
> > > >
> > > > Where u and v are nodes, and v_theOther_2u and u_theOther_2v are nodes connecting to u but not v, and connecting to v but not u, respectively. Appendix M provides an example illustrating how the LLM uses both p-values and domain knowledge to reason among these options. We have updated Appendix M to reference Appendix L2.
> > > >
> > > > >Regarding W1, if the LLM is imperfect, how does it help in the first place? Without assuming certain level of correctness from LLMs, how do we guarantee the correctness of the final results that depend on the LLMs outputs.
> > > >
> > > > Mmpx-R1-Q4: W1.
> > > >
> > > > Thank you for the insightful comment. In our framework, we do not assume that the LLM has any predictable accuracy, but rather that inaccuracies can be explicitly filtered out. MosaCD relies on the fact that the full LLM voting procedure (Step 2, which includes shuffling answer labels and performing a majority vote across multiple LLM calls; not just simply prompting an LLM) produces a more reliable initialization than collider orientation. This claim is supported empirically: (i) LLM-based seeding exhibits substantially fewer errors than collider orientation (average false seeds 4.8% vs. PC 26.7%; Figure 1 of the original submission), (ii) This robustness persists even when variable descriptions are uninformative (right panel in Figure 2 of the original submission), and (iii) stronger empirical performance of MosaCD (Table 1 of the original submission). Unlike prior work that assumes a fixed small LLM error rate, our domain-specific querying procedure explicitly reduces variance and bias through repeated sampling and majority voting, resulting in a more stable and accurate prior. We have updated Section 2 (lines 122-123) to clarify this.

---

> > > > > ### Comment · Reviewer_Mmpx · 2025-11-27
> > > > > **Thank you for the response**
> > > > >
> > > > > Thank you for the response. Regarding W3, what are the implications of updated line 5 of the algorithm? How does it affect the theory/guarantees? It is required to re-run the experiments as it is the crucial part of the algorithm?

---

> > > > > > ### Author Response · Authors · 2025-11-27
> > > > > >
> > > > > > Thank you for the follow-up question. We have re-examined both the theory and the implementation and confirmed that this was solely a typographical error: the algorithm and all analyses were already using the intended definition (i.e., “semi-directed path” should have been “directed path”). Therefore, correcting this typo does not affect the theoretical guarantees, the algorithmic behavior, or the experimental results. We apologize for the oversight and thank the reviewer again for pointing it out.

---

> > > > > > > ### Comment · Reviewer_Mmpx · 2025-11-27
> > > > > > > **Thank you for the response**
> > > > > > >
> > > > > > > Thank you for the discussion. The authors have answered my questions, but some responses appear insufficiently grounded in theory. In particular, the claims about “LLMs using quantitative cues” and “LLM has predictable accuracy” are vague and would benefit from clearer definitions or supporting references. I updated my score accordingly.

---

> ### Author Response · Authors · 2025-12-01
> **Thank you for the continued discussion**
>
> Mmpx-R2
>
> Thank you for the follow-up comment. We appreciate the reviewer’s concern and have expanded our original response to Mmpx-W1 to provide clearer definitions and supporting references for our statements regarding “LLMs using quantitative cues” and “LLM has predictable accuracy.”
>
> Specifically, we clarify that “existing imperfect-expert approaches typically rely on strong or unrealistic assumptions about LLM accuracy”, and summarize the supporting references (with quotes taken from the respective papers) below:
>
> - Long et al. assumes “an expert that returns correct orientations with some fixed probability”, thus requiring consistent levels LLM knowledge for every dataset. In contrast our method can handle any level of LLM error, even 100%, as shown in the null information setting of Figure 2.
> - Vashistha et al.  assumes “the $\epsilon$-expert to have error probability exactly equal to $\epsilon$”, with $\epsilon$ being small, and given a question with 3 options that “the probability for the two wrong options are equally likely”, thus requiring a small and predictable error rate for the queried questions. In contrast our method can handle any level of LLM error, even 100%, as shown in the null information setting of Figure 2.
> - Ban et al. relies on manually setting a “confidence threshold” which can vary greatly in prior accuracy at different thresholds for different datasets and different LLMs, as reported in their paper. In contrast our approach is shown to be robust across LLMs and does not require any confidence tuning to achieve reliable results.
>
> To clarify these points, we have added a new section comparing all LLM-based methods evaluated in our paper together with the assumptions each method makes. This table has been added to Appendix N.

---

### Official Review · Reviewer_g17M · 2025-11-01

**Soundness:** 3
**Presentation:** 3
**Contribution:** 2
**Rating:** 4
**Confidence:** 4

**Summary:**

** Summary

The paper proposes a hybrid causal discovery method that combines constraint-based methods and LLM-based methods, where the authors claim the potential to mitigate the error of collider-first order in traditional constraint-based methods. The proposed method, termed MosaCD, first initialises a skeleton by the same process as PC, then, orients high-confidence edges based on LLMs’ generation, and lastly, determines the other edges’ directions by propagation principles, conditional independence test results, and LLMs’ generations. The paper provides theoretical analysis on the correctness of MosaCD, and the error rates of traditional constraint-based methods. Lastly, it provides experimental analysis to show MosaCD’s efficacy.

** Recommendation

I would like to recommend a weak rejection to this paper as I am unclear about whether their method efficiently addresses the collider-first error. The paper raises an interesting problem of the traditional constraint-based causal discovery methods, and provides a formal proof, though it is based on strong assumptions. However, I am not clear whether the method can largely solve the collider-first issue because it basically uses the same conditional independence test results as PC, which means for each pair of nodes, there may exist at most one sepset. I am happy to increase my score if the authors clarify I am wrong on the following comments.

**Strengths:**

1. The paper studies the collider-first error of the traditional constraint-based methods, and provides a proof for that.
2. The proposed method tries to mitigate the error with some special design.

**Weaknesses:**

1. My main concern is that the method may fail to mitigate the claimed issue. Basically, since the sepsets are obtained by the same process as PC, each node pair may have at most one sepset. This means that MosaCD orients a v-structure is the middle node is not in this sepset. Then, this has the same problem. I understand that LLMs are used first, and this may reduce the error, however, LLMs can also bring much noise for its bad performance in causal reasoning.
2. The assumptions, e.g., 5.3 and 5.4, are quite strong. For instance, 5.3 is close to an asymptotic assumption, however, asymptotically, PC is guaranteed to orient correct v-structures. 5.4 then largely simplifies the graph.
3. Bnlearn benchmarks may not be good choices for evaluation because in general, LLMs have good memory on them. This undermines the convincingness of the work. Additionally, the authors may consider to compare to more recent LLM-based or hybrid methods.

**Questions:**

Questions see the weakness section.

** Minor comments

1. Define E_{seed} in Line 267
2. Line 156: dependence -> independence?

---

> ### Author Response · Authors · 2025-11-24
> **Response to Reviewer g17M (1/3)**
>
> Thank you for the accurate summary of our method and noting our theoretical and empirical results addressing the collider-orientation issues of PC. We summarize each review comment and provide a point-by-point response below. The manuscript has been revised accordingly, incorporating 2 new analyses, 2 updated figures/tables, and 2 new or updated supplementary figures/tables. We have also included 2 more small LLMs and 8 new baseline methods to ensure rigorous empirical evaluation.
>
> The reviewer has raised 3 comments/weaknesses/questions. We believe that we have completely addressed 2 out of these 3 points (W1,W3) and substantially addressed the remaining points (W2). Please let us know if there are any additional questions or if further clarification would be helpful.
>
> >My main concern is that the method may fail to mitigate the claimed issue. Basically, since the sepsets are obtained by the same process as PC, each node pair may have at most one sepset. This means that MosaCD orients a v-structure is the middle node is not in this sepset. Then, this has the same problem. I understand that LLMs are used first, and this may reduce the error, however, LLMs can also bring much noise for its bad performance in causal reasoning.
>
> g17M-W1: If MosaCD can mitigate the collider-first error.
>
> The reviewer is correct that MosaCD uses the same v-structure criterion as PC-style algorithms (Step 3.3) and therefore has identical accuracy for determining colliders when this criterion is used. The key distinction lies in when collider orientation is used. PC-style methods apply collider orientation at the very beginning for seeding directions. In contrast, MosaCD applies collider orientation only at a much later stage: after LLM-based orientation seeding (Step 2) and after other reliable propagation steps (Steps 3.1–3.2). This design mitigates the collider-first error because: (i) LLM-based seeding is substantially more accurate than collider-based seeding, and (ii) collider orientation is applied only after most edges have already been oriented, meaning it affects a much smaller subset of edges and thus cannot propagate early collider errors as PC does. We elaborate on both points below.
>
> (i) LLM-based seeding is substantially more accurate than collider-based seeding.
> PC-style algorithms rely on collider orientation to generate initial directions, whereas MosaCD seeds orientations using LLM. As shown in Figure 1 of the original submission, LLM-based seeding (“MosaCD”) achieves significantly higher accuracy than collider-based seeding (“PC variants”), with a higher amount of true seeds in 7/10 datasets and far fewer false seeds overall (4.8% vs. 26.7%). In addition, Figure 2 shows that MosaCD remains robust even when variable descriptions are uninformative. These results indicate that LLM-based seeding provides a more reliable starting point and reduces the risk of early error propagation.
>
> (ii) Collider orientation is used on far fewer edges in MosaCD.
> We performed additional analyses to quantify how often collider orientation determines an edge direction. Across the 10 datasets, MosaCD uses collider orientation on far fewer edges than PC (average 3.3 for MosaCD vs. 28.5 for PC variants). This demonstrates that MosaCD depends much less on collider-based orientation, limiting the negative impact of collider-first errors.
>
> We have updated the manuscript to clarify these points and report the new analyses Section 6.1 (lines 417-418).

---

> ### Author Response · Authors · 2025-11-24
> **Response to Reviewer g17M (2/3)**
>
> >The assumptions, e.g., 5.3 and 5.4, are quite strong. For instance, 5.3 is close to an asymptotic assumption, however, asymptotically, PC is guaranteed to orient correct v-structures. 5.4 then largely simplifies the graph.
>
> g17M-W2: Justification for assumptions 5.3 and 5.4.
>
> Thank you for the comment. We recognize the importance of clearly justifying Assumptions 5.3 and 5.4. These assumptions are introduced solely to make the theoretical analysis tractable; they are not required for running MosaCD in practice.
>
> Assumption 5.3 is intended to model the fact that CI tests are imperfect. Our analysis considers an asymptotic regime in which the number of neighboring nodes M=|V\{X,Y}| grows (rather than the sample size), and the CI error rates $\alpha,\beta$ decreases with M. Theorem 5.5 applies to any finite M where with nonzero CI error$ \alpha,\beta >0$, where the classical PC is not guaranteed to orient v-structures correctly, which assumes oracle-perfect CI tests. Assumption 5.3 therefore allows us to study error propagation in a realistic, non-oracle setting, rather than in the idealized asymptotic setting like in the PC paper (analogous to Assumption 5.1 in our manuscript).
>
> Assumption 5.4 imposes a locally sparse graph structure without redundant X—Y paths. This reflects that in many real-world scenarios where each variable depends on only a small subset of others and, as the reviewer notes.  This assumption is made to keep the combinatorial analysis manageable. It is a structural simplification for the theoretical model, not an operational requirement of the algorithm.
>
> Importantly, we explicitly treat Assumptions 5.3–5.4 as simplifying assumptions for a toy model that isolates the core mechanism of error propagation we study. Appendix K demonstrates that the qualitative conclusion—namely, that non-collider-first ordering yields lower error than collider-first ordering—continues to hold even when these assumptions are relaxed. We have revised the manuscript Section 5.1 (lines 307–308) to clarify the purpose and limitations of these assumptions.
>
>
> >Bnlearn benchmarks may not be good choices for evaluation because in general, LLMs have good memory on them. This undermines the convincingness of the work. Additionally, the authors may consider to compare to more recent LLM-based or hybrid methods.
>
> g17M-W3: LLM memorization and additional baselines.
>
> Thank you for this comment. The authors raised 2 related points, which we answer in turn.
>
> LLM Memorization: To address your concern, we provide 2 results supporting that our empirical results are not driven by memorization.
>
> We applied our method and baselines to 2 new datasets from [5] that were determined to be unaffected by LLM memorization, namely COVID-19 Respiratory (11 nodes) and COVID-19 Complications (63 nodes). As shown in the table below, MosaCD continues to outperform PC and Meek, indicating that it is robust to lower information settings, consistent with our null information simulations in Figure 2. We’ve added these results to Section 6.1 (lines 378-379) and Appendix E.
>
> | Method     | Covid Respiratory (11) | Covid Complications (63) |
> | ---------- | ----------------- | ------------------- |
> | **PC**     | 0.31              | 0.24                |
> | **Meek**   | 0.32              | 0.29                |
> | **MosaCD** | 0.52              | 0.30                |
> *F1 scores across methods and datatsets*
>
> We added 4 more LLM additional baselines to benchmark: (i) Vashishtha [2] (prompted with the entire graph), (ii) Jiralerspong [3] (prompted on edge at a time) are oriented almost completely through the LLM alone, (iii) Causal Disco [1], which uses an imperfect LLM prior, (iv) LLM-CD [4], which also uses an imperfect LLM prior. Vashishtha and Jiralerspong are prompted in a manner similar to [5]’s memorization stress tests; therefore, if LLM memorization were driving performance, these methods would be expected to perform very well. However, as shown in the table at the beginning, MosaCD outperforms all LLM-based methods on 7 out of 10 datasets, often by a large margin—for example, achieving up to 0.33 and 0.80 higher F1 scores than Vashishtha and Jiralerspong, respectively, on the Child dataset. These results indicate that MosaCD takes the best advantage of available LLM knowledge and its performance gains are unlikely to be attributable to LLM memorization. We have updated Table 1 in Section 6.1 (lines 370-380) to report these findings.
>
>
> [1] Long, Stephanie, et al. "Causal discovery with language models as imperfect experts."
>
> [2] Causal Order: The Key to Leveraging Imperfect Experts in Causal Inference (Vashishtha et al.,2025)
>
> [3] Efficient Causal Graph Discovery Using Large Language Models (Jiralerspong et al., 2024)
>
> [4] Integrating Large Language Model for Improved Causal Discovery (Ban et al., 2025)
>
> [5] Realizing LLMs’ Causal Potential Requires Science-Grounded, Novel Benchmarks (Srivastava et al., 2025)

---

> ### Author Response · Authors · 2025-11-24
> **Response to Reviewer g17M (3/3)**
>
> g17M-W3 continued:
>
> Please see our updated main results table (Table 1) below with the 8 additional baselines. (F1 scores across datasets).
>
> | Method             | Cancer (5) | Asia (8) | Child (20)  | Insurance (27) | Water (32)  |
> | ------------------ | ---------- | -------- | ----------- | -------------- | ----------- |
> | **PC**             | 0.50       | 0.67     | 0.70        | 0.62           | 0.45        |
> | **Meek**           | 0.50       | 0.67     | 0.78        | 0.70    | 0.57        |
> | **Shapley-PC**     | **1.00**   | 0.53     | 0.67        | 0.67           | 0.47        |
> | **ILS-CSL***       | 0.50       | **0.93** | 0.83| 0.70    |0.60 |
> | **SCP***           | 0.50       | 0.67     | 0.78        | 0.68           | 0.57        |
> | **Jiralerspong***  | 0.57       | 0.55     | 0.10        | 0.21           | 0.00        |
> | **Vashishtha***    | 0.75       | 0.93     | 0.67        | 0.70    | 0.65        |
> | **Causal Disco***  | 0.57       | 0.86     | 0.56        | 0.34           | **0.87**    |
> | **LLM-CD***        | **1.00**   | 0.80     | 0.40        | 0.29           | 0.24        |
> | **NOTEARS**        | 0.00       | 0.57     | 0.48        | 0.15           | 0.29        |
> | **DAG-GNN**        | 0.00       | 0.27     | 0.24        | 0.20           | 0.26        |
> | **GES**            | 0.50       | 0.12     | 0.17        | 0.26           | 0.29        |
> | **HC**             | 0.50       | 0.12     | 0.33        | 0.33           | 0.36        |
> | **MosaCD*** (Ours) | **1.00**   | **0.93** | **0.90**    | **0.87**       | 0.59        |
>
> | Method             | Mildew (35) | Alarm (37)  | Hailfinder (56) | Hepar2 (70) | Win95pts (76) |
> | ------------------ | ----------- | ----------- | --------------- | ----------- | ------------- |
> | **PC**             | 0.63        | 0.85        | 0.38            | 0.36        | 0.59          |
> | **Meek**           | 0.69        | 0.90 | 0.40            | 0.39        | 0.64          |
> | **Shapley-PC**     | 0.75        | 0.84        | 0.38            | 0.44        | 0.65          |
> | **ILS-CSL***       | 0.89 | 0.85        | 0.44            | 0.54        | 0.69          |
> | **SCP***           | 0.69        | 0.87        | 0.39            | 0.38        | 0.63          |
> | **Jiralerspong***  | 0.07        | 0.09        | 0.10            | 0.16        | 0.03          |
> | **Vashishtha***    | 0.78        | 0.72        | **0.60**        | 0.66        | 0.74   |
> | **Causal Disco***  | 0.61        | 0.51        | 0.33            | **0.81**    | 0.55          |
> | **LLM-CD***        | 0.24        | 0.30        | 0.22            | 0.32        | 0.17          |
> | **NOTEARS**        | 0.25        | 0.25        | 0.27            | 0.06        | 0.14          |
> | **DAG-GNN**        | 0.43        | 0.53        | 0.13            | 0.13        | 0.37          |
> | **GES**            | 0.23        | 0.26        | 0.33            | 0.45        | 0.51          |
> | **HC**             | 0.24        | 0.30        | 0.31            | 0.47        | 0.46          |
> | **MosaCD*** (Ours) | **0.90**    | **0.93**    | 0.49    | 0.72 | **0.81**      |
> *F1 scores across datasets and methods*
>
> More LLM or hybrid methods: We add 8 additional baselines. Including the 4 more LLM methods mentioned above, we add continuous optimization methods NOTEARS and DAG-GNN and score-based methods GES and HC. See table above for results. MosaCD outperforms all methods in 7/10 datasets and the strongest performance overall (average F1 score 0.81 versus next best 0.69).

---

### Official Review · Reviewer_JtHM · 2025-11-02

**Soundness:** 2
**Presentation:** 3
**Contribution:** 2
**Rating:** 4
**Confidence:** 4

**Summary:**

This work introduces MosaCD, a new causal discovery algorithm that integrates traditional constraint-based methods (like PC, CPC, and PC-Stable) with LLM priors to improve the accuracy and robustness of learning causal graphs from observational data. Classical approaches rely heavily on conditional independence (CI) tests and v-structure orientations, which are error-prone with finite samples and can cause cascading mistakes. MosaCD mitigates these issues by using LLMs to propose initial edge orientations, while filtering out hallucinations via shuffled multiple choice queries that exploit positional bias, keeping only consistent, high-confidence edges. It further introduces a confidence-prioritized propagation strategy that orients non-colliders before colliders, shown theoretically and empirically to reduce errors. The work seems promising and well grounded, however it can benefit from more empirical analysis, given high potential of data leakage on the benchmarks the paper evaluates on.

**Strengths:**

1. The paper makes a clear conceptual leap by embedding LLMs directly inside the causal discovery pipeline, rather than using them only as post-hoc annotators or priors.

2. The new confidence-down propagation rule orienting edges with higher CI confidence first and prioritizing non-collider identification over colliders is theoretically grounded and empirically validated.

3. The approach is modular and model-agnostic, thus enabling generalizable adoption.

4. Paper shows strong empirical improvements.

**Weaknesses:**

1. Shuffling order and re-querying is a well known way of averaging out positional bias which has been used in many prior papers [1]. So i feel the claim that the proposed framework removes hallucination is heavily overloaded and lacks novelty.

2. The datasets on which the framework is evaluated belongs from the well known BNLearn benchmark, with a very high chance of leakage during model's training. Recent studies like [2] confirm that LLMs have memorized the datasets from BNLearn that this study uses for evaluation. *I would strongly suggest tackling this claim*.

3. Computational overhead from repeated LLM queries due to shuffling of query order seems like a big issue which is not properly addressed. This will become significantly higher when costly reasoning models would be used for inference for large causal graphs.

4. Although MosaCD outperforms PC variants and recent LLM-based methods (ILS-CSL, SCP), it does not compare against hybrid or differentiable structure learning approaches (e.g., NOTEARS, DAG-GNN) that could also integrate priors. Thus, it remains unclear how its benefits scale relative to more flexible score-based frameworks. Also lack of comparison with recent work on integrating LLMs with traditional discovery methods [3][4][5] is a problem, since this paper proposes lacks comparison with recently proposed frameworks and fails to highlight how it stands out in comparison to those work.

5. F1 score is a limited metric, especially when dealing with structural evaluation of a graph. Try incorporating Structural Hamming Distance and Topological Divergence [3] which are more graph specific metrics and relate to real world causal inference tasks like effect estimation.

6. Qualitative analysis of the LLM reasoning behavior will help understand the impact of the question framing effect, which is not provided currently.

7. The paper emphasizes on PC, whose performance is heavily conditioned on the provided sample size. I am curious if their framework will help get significant improvement under scarce data settings, especially when the data samples are very low (<1,000).

8. Please incorporate more for evaluation, as current analysis only limits to GPT-4o-mini. Try to incorporate open source models from Qwen, Llama family for wider adoption of your framework.

References:
[1] Large Language Models Sensitivity to The Order of Options in Multiple-Choice Questions (Pezeshkpour et al., 2023)

[2] Realizing LLMs’ Causal Potential Requires Science-Grounded, Novel Benchmarks (Srivastava et al., 2025)

[3] Causal Order: The Key to Leveraging Imperfect Experts in Causal Inference (Vashishtha et al.,2025)

[4] Efficient Causal Graph Discovery Using Large Language Models (Jiralerspong et al., 2024)

[5] Causal Structure Learning Supervised by Large Language Model (Ban et al., 2023)

**Questions:**

Please try to answer each point enlisted under weaknesses. I believe the work has potential, but lacks in empirical rigor. I'll be happy to increase my scores if the author can show proof and tackle all points I mentioned under weaknesses. I believe the framework's effectiveness cannot be judged until the data leakage issue is resolved, since recent papers show that the graphs used for evaluation in this work from BNLearn are memorized, thus potentially skewing the results heavily.

---

> ### Author Response · Authors · 2025-11-24
> **Response to Reviewer JtHM (1/5)**
>
> Thank you for the thoughtful summary of our work and noting our theoretical and empirical results addressing the collider-orientation issues of PC. We summarize each review comment and provide a point-by-point response below. The manuscript has been revised accordingly, incorporating 2 new analyses, 2 updated figures/tables, and 2 new or updated supplementary figures/tables. We have also included 2 more small LLMs and 8 new baseline methods to ensure rigorous empirical evaluation.
>
> The reviewer has raised 8 comments/weaknesses/questions. We believe that we have completely addressed 7 out of these 8 points (W1,W2,W3,W4,W5,W7,W8) and substantially addressed the remaining points (W6). Please let us know if there are any additional questions or if further clarification would be helpful.
>
> Please see our updated main results table (Table 1) below with the 8 additional baselines. (F1 scores across datasets).
>
> | Method             | Cancer (5) | Asia (8) | Child (20)  | Insurance (27) | Water (32)  |
> | ------------------ | ---------- | -------- | ----------- | -------------- | ----------- |
> | **PC**             | 0.50       | 0.67     | 0.70        | 0.62           | 0.45        |
> | **Meek**           | 0.50       | 0.67     | 0.78        | 0.70    | 0.57        |
> | **Shapley-PC**     | **1.00**   | 0.53     | 0.67        | 0.67           | 0.47        |
> | **ILS-CSL***       | 0.50       | **0.93** | 0.83| 0.70    |0.60 |
> | **SCP***           | 0.50       | 0.67     | 0.78        | 0.68           | 0.57        |
> | **Jiralerspong***  | 0.57       | 0.55     | 0.10        | 0.21           | 0.00        |
> | **Vashishtha***    | 0.75       | 0.93     | 0.67        | 0.70    | 0.65        |
> | **Causal Disco***  | 0.57       | 0.86     | 0.56        | 0.34           | **0.87**    |
> | **LLM-CD***        | **1.00**   | 0.80     | 0.40        | 0.29           | 0.24        |
> | **NOTEARS**        | 0.00       | 0.57     | 0.48        | 0.15           | 0.29        |
> | **DAG-GNN**        | 0.00       | 0.27     | 0.24        | 0.20           | 0.26        |
> | **GES**            | 0.50       | 0.12     | 0.17        | 0.26           | 0.29        |
> | **HC**             | 0.50       | 0.12     | 0.33        | 0.33           | 0.36        |
> | **MosaCD*** (Ours) | **1.00**   | **0.93** | **0.90**    | **0.87**       | 0.59        |
>
> | Method             | Mildew (35) | Alarm (37)  | Hailfinder (56) | Hepar2 (70) | Win95pts (76) |
> | ------------------ | ----------- | ----------- | --------------- | ----------- | ------------- |
> | **PC**             | 0.63        | 0.85        | 0.38            | 0.36        | 0.59          |
> | **Meek**           | 0.69        | 0.90 | 0.40            | 0.39        | 0.64          |
> | **Shapley-PC**     | 0.75        | 0.84        | 0.38            | 0.44        | 0.65          |
> | **ILS-CSL***       | 0.89 | 0.85        | 0.44            | 0.54        | 0.69          |
> | **SCP***           | 0.69        | 0.87        | 0.39            | 0.38        | 0.63          |
> | **Jiralerspong***  | 0.07        | 0.09        | 0.10            | 0.16        | 0.03          |
> | **Vashishtha***    | 0.78        | 0.72        | **0.60**        | 0.66        | 0.74   |
> | **Causal Disco***  | 0.61        | 0.51        | 0.33            | **0.81**    | 0.55          |
> | **LLM-CD***        | 0.24        | 0.30        | 0.22            | 0.32        | 0.17          |
> | **NOTEARS**        | 0.25        | 0.25        | 0.27            | 0.06        | 0.14          |
> | **DAG-GNN**        | 0.43        | 0.53        | 0.13            | 0.13        | 0.37          |
> | **GES**            | 0.23        | 0.26        | 0.33            | 0.45        | 0.51          |
> | **HC**             | 0.24        | 0.30        | 0.31            | 0.47        | 0.46          |
> | **MosaCD*** (Ours) | **0.90**    | **0.93**    | 0.49    | 0.72 | **0.81**      |
> *SHD scores across datasets and methods*

---

> ### Author Response · Authors · 2025-11-24
> **Response to Reviewer JtHM (2/5)**
>
> > Shuffling order and re-querying is a well known way of averaging out positional bias which has been used in many prior papers [1]. So i feel the claim that the proposed framework removes hallucination is heavily overloaded and lacks novelty.
>
> JtHM-W1: Novelty of removing hallucination by shuffling order.
>
> Thank you for this comment. We agree that shuffling answer order and re-querying to mitigate positional bias has been used in the LLM literature [1], and we do not claim to have introduced a new LLM mechanism. Our contribution lies in adapting and tailoring this idea to the causal discovery setting. In particular, we designed a domain-specific querying strategy for proposing edge directions, and showed that it can be used to reliably seed a causal graph. This is novel to the best of our knowledge. We have updated claim 2 in Section 1 (lines 79-80) to clarify this.
>
> “We demonstrate how LLMs can be adapted for robust seeding in causal discovery with a domain-specific querying strategy to minimize hallucination influence.”
>
> [1] Large Language Models Sensitivity to The Order of Options in Multiple-Choice Questions (Pezeshkpour et al., 2023)
>
> > The datasets on which the framework is evaluated belongs from the well known BNLearn benchmark, with a very high chance of leakage during model's training. Recent studies like [2] confirm that LLMs have memorized the datasets from BNLearn that this study uses for evaluation. I would strongly suggest tackling this claim.
>
> JtHM-W2: LLM memorization of dataset.
>
> Thank you for this comment. It is indeed important to evaluate the impact of LLM memorization on empirical performance. The paper you mentioned shows some level of memorization on 5/10 of our datasets. To address your concern, we provide 2 results supporting that our empirical results are not driven by memorization.
>
> (1) We applied our method and baselines to 2 new datasets from [2] that were determined to be unaffected by LLM memorization, namely COVID-19 Respiratory (11 nodes) and COVID-19 Complications (63 nodes). As shown in the table below, MosaCD continues to outperform PC and Meek, indicating that it is robust to lower information settings, consistent with our null information simulations in Figure 2. We’ve added these results to Section 6.1 (lines 378-379) and Appendix E.
>
> | Method     | Covid Respiratory (11) | Covid Complications (63) |
> | ---------- | ----------------- | ------------------- |
> | **PC**     | 0.31              | 0.24                |
> | **Meek**   | 0.32              | 0.29                |
> | **MosaCD** | 0.52              | 0.30                |
> *F1 scores across datasets and methods*
>
> [2] Realizing LLMs’ Causal Potential Requires Science-Grounded, Novel Benchmarks (Srivastava et al., 2025)
>
> (2) We added 4 more LLM additional baselines to benchmark: (i) Vashishtha [4] (prompted with the entire graph), (ii) Jiralerspong [5] (prompted on edge at a time) are oriented almost completely through the LLM alone, (iii) Causal Disco [5], which uses an imperfect LLM prior, (iv) LLM-CD [6], which also uses an imperfect LLM prior. Vashishtha and Jiralerspong are prompted in a manner similar to [2]'s memorization stress tests; therefore, if LLM memorization were driving performance, these methods would be expected to perform very well. However, as shown in the table at the beginning, MosaCD outperforms all LLM-based methods on 7 out of 10 datasets, often by a large margin—for example, achieving up to 0.33 and 0.80 higher F1 scores than Vashishtha and Jiralerspong, respectively, on the Child dataset. These results indicate that MosaCD takes the best advantage of available LLM knowledge and its performance gains are unlikely to be attributable to LLM memorization. We have updated Table 1 in Section 6.1 (lines 371-380) to report these findings.
>
> [3] Long, Stephanie, et al. "Causal discovery with language models as imperfect experts."
>
> [4] Causal Order: The Key to Leveraging Imperfect Experts in Causal Inference (Vashishtha et al.,2025)
>
> [5] Efficient Causal Graph Discovery Using Large Language Models (Jiralerspong et al., 2024)
>
> [6] Integrating Large Language Model for Improved Causal Discovery (Ban et al., 2025)

---

> ### Author Response · Authors · 2025-11-24
> **Response to Reviewer JtHM (3/5)**
>
> >  Computational overhead from repeated LLM queries due to shuffling of query order seems like a big issue which is not properly addressed. This will become significantly higher when costly reasoning models would be used for inference for large causal graphs.
>
> JtHM-W3: Computational overhead.
>
>
> Thank you for the question. We have now measured and reported the computational cost of the shuffle-and-vote procedure and the total CI testing time across 3 datasets used in our ablation studies (“Asia,” “Insurance,” “Hepar2”) and 3 skeleton methods (“PC,” “PC-Stable,” “CPC”). Runtime in seconds is reported below and added to Appendix C. We find that the shuffle-and-vote step takes longer than the CI tests themselves (avg. ~10x  longer for Hepar2). Although this overhead is non-negligible, we find it acceptable in practice. Accessing a commercial model through their API may mitigate this, as parallel prompting would then be easily achievable. We have updated Section 6.1 (lines 421-422) and Appendix F to report these results. We also note that as per Figure 4, costly reasoning models may not be needed, as low latency GPT-4o-mini gets the same performance as reasoning model GPT-5. We acknowledge an efficient prompting method is an important future direction to maximize runtime efficiency.
>
>
> | Method          | Asia PC | Asia PC-Stable | Asia CPC | Insurance PC | Insurance PC-Stable | Insurance CPC | Hepar2 PC | Hepar2 PC-Stable | Hepar2 CPC |
> | --------------- | ------- | -------------- | -------- | ------------ | ------------------- | ------------- | --------- | ---------------- | ---------- |
> | **Skeleton**    | 0.23    | 0.50           | 0.55     | 50.32        | 76.51               | 75.89         | 187.05    | 313.21           | 264.50     |
> | **LLM**         | 256.82  | 232.13         | 264.14   | 1501.94      | 1397.03             | 1434.65       | 2492.51   | 2453.28          | 2260.18    |
> | **Propagation** | 0.00    | 0.00           | 0.00     | 0.01         | 0.01                | 0.01          | 0.03      | 0.03             | 0.03       |
> *Runtime (seconds) for each step across datasets and skeletons*
>
> > Although MosaCD outperforms PC variants and recent LLM-based methods (ILS-CSL, SCP), it does not compare against hybrid or differentiable structure learning approaches (e.g., NOTEARS, DAG-GNN) that could also integrate priors. Thus, it remains unclear how its benefits scale relative to more flexible score-based frameworks. Also lack of comparison with recent work on integrating LLMs with traditional discovery methods [3][4][5] is a problem, since this paper proposes lacks comparison with recently proposed frameworks and fails to highlight how it stands out in comparison to those work.
>
> JtHM-W4: Baselines.
>
> Thank you for the comment. We agree that a comprehensive comparison is important. We have added 8 new baselines to our evaluation, including 4 requested by the reviewer (NOTEARS, DAG-GNN, Vashistha et al.[2], Jiralerspong et al. [3]), two additional LLM methods (Causal Disco[1], LLM-CD[4]) and 2 additional score-based methods (GES and HC). We note that the other baseline requested by the reviewer, ILS-CSL[5], was already included in the original submission. Results for these expanded baselines are reported in Table 1 above. MosaCD achieves top performance on 7/10 tasks relative to these 8 new baselines and the best performance overall (average MosaCD F1 score 0.81 versus next best 0.69), demonstrating its superior performance. We have updated Section 6.1 (lines 371-380) to report these findings.
>
> [1] Long, Stephanie, et al. "Causal discovery with language models as imperfect experts."
>
> [2] Causal Order: The Key to Leveraging Imperfect Experts in Causal Inference (Vashishtha et al.,2025)
>
> [3] Efficient Causal Graph Discovery Using Large Language Models (Jiralerspong et al., 2024)
>
> [4] Integrating Large Language Model for Improved Causal Discovery (Ban et al., 2025)
>
> [5] Causal Structure Learning Supervised by Large Language Model (Ban et al., 2023)

---

> ### Author Response · Authors · 2025-11-24
> **Response to Reviewer JtHM (4/5)**
>
> > F1 score is a limited metric, especially when dealing with structural evaluation of a graph. Try incorporating Structural Hamming Distance and Topological Divergence [3] which are more graph specific metrics and relate to real world causal inference tasks like effect estimation.
>
> JtHM-W5: Additional evaluation metrics.
>
> Thank you for this comment. We agree that including additional metrics provides a more comprehensive assessment. In response, we have added Structural Hamming Distance (SHD) to Appendix B. SHD leads to similar qualitative conclusions as the F1 metric: MosaCD continues to outperform the existing baselines in the same 7/10 datasets. We have updated the manuscript accordingly (lines 375-376).
>
> | Method             | Cancer (5) | Asia (8) | Child (20) | Insurance (27) | Water (32) |
> | ------------------ | ---------- | -------- | ---------- | -------------- | ---------- |
> | **PC**             | 2.0        | 3.0      | 10.5       | 28.0           | 53.0       |
> | **Meek**           | 2.0        | 3.0      | 6.0        | 19.0           | 38.0       |
> | **Shapley-PC**     | **0.0**    | 4.0      | 19.0       | 27.5           | 52.5       |
> | **ILS-CSL***       | 2.0        | **1.0**  | 5.5        | 25.0           | 37.5       |
> | **SCP***           | 2.0        | 3.0      | 6.0        | 21.0           | 38.0       |
> | **Jiralerspong***  | 3.0        | 5.0      | 38.0       | 52.0           | 71.0       |
> | **Vashishtha***    | 1.0        | **1.0**  | 9.0        | 21.0           | 34.0       |
> | **Causal Disco***  | 2.0        | 2.0      | 13.0       | 41.0           | **15.0**   |
> | **LLM-CD***        | **0.0**        | 2.0      | 30.0       | 59.0           | 81.0       |
> | **DAG-GNN**        | 4.0        | 9.0      | 23.0       | 60.0           | 55.0       |
> | **GES**            | 2.0        | 9.0      | 63.0       | 105.0          | 71.0       |
> | **HC**             | 2.0        | 9.0      | 51.0       | 95.0           | 60.0       |
> | **MosaCD*** (Ours) | **0.0**    | **1.0**  | **2.0**    | **12.0**       | 36.0       |
>
> | Method             | Mildew (35) | Alarm (37) | Hailfinder (56) | Hepar2 (70) | Win95pts (76) |
> | ------------------ | ----------- | ---------- | --------------- | ----------- | ------------- |
> | **PC**             | 23.0        | 12.0       | 105.0           | 99.5        | 68.0          |
> | **Meek**           | 17.0        | 7.5        | 97.0            | 87.0        | 57.5          |
> | **Shapley-PC**     | 18.5        | 14.0       | 105.0           | 89.0        | 69.5          |
> | **ILS-CSL***       | 10.0        | 12.0       | 96.5            | 77.5        | 54.0          |
> | **SCP***           | 17.0        | 8.0        | 99.0            | 88.0        | 56.0          |
> | **Jiralerspong***  | 50.0        | 80.0       | 72.0            | 137.0       | 135.0         |
> | **Vashishtha***    | 13.0        | 15.0       | 77.0            | 59.0        | 46.0          |
> | **Causal Disco***  | 24.0        | 27.0       | **53.0**        | **36.0**    | 69.0          |
> | **LLM-CD***        | 52.0        | 59.0       | 92.0            | 117.0       | 160.0         |
> | **DAG-GNN**        | 50.0        | 35.0       | 77.0            | 119.0       | 93.0          |
> | **GES**            | 152.0       | 101.0      | 150.0           | 122.0       | 122.0         |
> | **HC**             | 140.0       | 92.0       | 162.0           | 119.0       | 133.0         |
> | **MosaCD*** (Ours) | **8.0**     | **6.0**    | 90.0            | 52.0        | **37.0**      |
> *SHD scores across datasets and methods*
>
> >Qualitative analysis of the LLM reasoning behavior will help understand the impact of the question framing effect, which is not provided currently.
>
> JtHM-W6: Qualitative analysis of the LLM reasoning behavior.
>
> Thank you for this comment. We conducted a qualitative analysis of the LLM outputs to better understand framing effects and reasoning patterns. Our analysis revealed several consistent behaviors: (i) a strong positional bias, with the first option selected disproportionately often across datasets (as in Appendix D); (ii) a reluctance to choose “I don’t know,” even when it is presented as the first option (as in Appendix D); and (iii) the LLM can use the provided p-values to reinforce opinions when it is not as sure (see example below). These observations help explain why our domain-specific querying strategy improves stability and performance. We have updated Appendix M to report these findings.
>
> “C and D contradict domain knowledge: dyspnoea is an effect, not a cause, of diseases like tuberculosis or lung cancer. The CI tests (p-values ~0.7–1) indicate no direct link between dyspnoea and the other variables given 'either', invalidating claims of dyspnoea influencing 'either' or its dependencies. ”

---

> ### Author Response · Authors · 2025-11-24
> **Response to Reviewer JtHM (5/5)**
>
> > The paper emphasizes on PC, whose performance is heavily conditioned on the provided sample size. I am curious if their framework will help get significant improvement under scarce data settings, especially when the data samples are very low (<1,000).
>
> JtHM-W7: Impact of sample size on performance.
>
> Thank you for the question. We agree that understanding performance under scarce data settings is important. As shown in Appendix F (Appendix D of the original submission), we evaluated the effect of sample size on the Insurance dataset by subsampling to different N. We find that MosaCD achieves consistently larger improvements over PC when the sample size is small, see F1 scores in the Insurance dataset in the table below. Notably, as the sample size becomes exceedingly small, MosaCD does not detect large amounts of false seeds (at most 2 versus PC 10). This indicates that our framework is beneficial to constraint-based discovery across any sample size regime. We have updated Section 6.1 (lines 421-422) and Appendix G to clarify this finding.
>
> | Method     | 100  | 1000 | 10000 |
> | ---------- | ---- | ---- | ----- |
> | **PC**     | 0.20 | 0.39 | 0.65  |
> | **Meek**   | 0.30 | 0.47 | 0.64  |
> | **MosaCD** | 0.37 | 0.67 | 0.84  |
> *F1 scores across sample sizes and methods*
>
> >Please incorporate more for evaluation, as current analysis only limits to GPT-4o-mini. Try to incorporate open source models from Qwen, Llama family for wider adoption of your framework.
>
> JtHM-W8: Incorporating more LLMs.
>
> Thank you for the suggestion. Section 6.2 of the original submission (Figure 4) already reported results across 5 LLMs, including the open-source GPT-oss-120B and several commercial models (GPT-4o-mini, GPT-5, Claude 3.5 Haiku, Claude 4 Sonnet). Following the reviewer’s recommendation, we have now expanded the evaluation to include two additional open-source models: Qwen-3 32B and Llama-3 8B-Instruct. The updated results (reported below) show that MosaCD continues to perform strongly with these smaller, locally runnable LLMs, for instance Qwen-3 32B gets the same performance as GPT-5, demonstrating that our framework is fully compatible with widely adoptable open-source alternatives. We have updated Figure 4 (lines 477-482) to report these findings.
>
> | Model                     | Asia | Insurance | Hepar2 |
> | ------------------------- | ---- | --------- | ------ |
> | **Claude 3.5 Haiku**      | 0.93 | 0.68      | 0.67   |
> | **Claude 4 Sonnet**       | 0.93 | 0.87      | 0.70   |
> | **GPT-4o-mini (Default)** | 0.93 | 0.87      | 0.72   |
> | **GPT-5**                 | 0.93 | 0.85      | 0.72   |
> | **Llama-3-8B**            | 0.93 | 0.56      | 0.59   |
> | **GPT-OSS-120B**          | 0.67 | 0.72      | 0.68   |
> | **Qwen3-32B**             | 0.93 | 0.87      | 0.71   |
> *MosaCD F1 scores across LLMs and datasets*

---

### Official Review · Reviewer_qqS8 · 2025-11-08

**Soundness:** 4
**Presentation:** 3
**Contribution:** 4
**Rating:** 8
**Confidence:** 4

**Summary:**

MosaCD is a constraint-based causal discovery algorithm that augments PC-style orientation with LLM-based seeding and a confidence-prioritized propagation strategy. It mitigates error cascades from imperfect CI tests by (1) using LLM queries to propose initial edge directions, (2) applying a shuffle-and-vote strategy to filter hallucinations, and (3) prioritizing orientations supported by reliable CI evidence—specifically orienting non-colliders first.

Theoretical analysis demonstrates that this reversal reduces orientation error rates compared to collider-first methods under noisy CI tests. Empirically, MosaCD achieves strong gains across 10 real and synthetic BNLearn graphs (5–76 nodes), outperforming PC, PC-Stable, CPC, and recent LLM-assisted methods (SCP, ILS-CSL). The method remains competitive even with noisy variable descriptions and across different LLM backbones.

**Strengths:**

- Conceptual elegant way of designing the algorithm and rethinking how LLMs should contribute to constraint-based discovery, avoiding naïve replacement.
-  Formal  analsys of soundness and an elegant asymptotic argument showing that non-collider orientation reduces expected errors.
- Rigorous empirical evaluation with 10 benchmarks, 3 skeleton variants, detailed ablations on hallucination filtering and seed robustness.
- Performance degrades gracefully under uninformative variables and different LLMs.
- Compatible with any PC-style skeleton learner.

**Weaknesses:**

- Dependence on large commercials models, no comparison to open-source smaller or distilled models presented.
- The confidence-down propagation weights are heuristic; a probabilistic interpretation or adaptive β schedule would strengthen robustness.
- Evaluation metrics relies on F1 of edge orientations only; additional causal sufficiency or equivalence-class metrics (e.g., SID, SHD) would provide a fuller view.

**Questions:**

1) What is the computational cost of the shuffle-and-vote procedure relative to the total CI test time?
2) Could confidence propagation be formalized probabilistically (e.g., Bayesian edge belief updating)?
3) How does MosaCD handle contradictory LLM votes or inconsistent priors across iterations?
4) Can the approach scale to 100–500 nodes or use learned embeddings to approximate CI filtering?
5) Can the method work with smaller or distilled language models that could be run locally?

---

> ### Author Response · Authors · 2025-11-24
> **Response to Reviewer qqs8 (1/3)**
>
> Thank you for the accurate summary of our method and for highlighting the rigor of our evaluation setup. We summarize each review comment and provide a point-by-point response below. The manuscript has been revised accordingly, incorporating 2 new analyses, 2 updated figures/tables, and 2 new or updated supplementary figures/tables. We have also included 2 more small LLMs and 8 new baseline methods to ensure rigorous empirical evaluation.
>
> The reviewer has raised 3 comments/weaknesses and 5 questions. We believe that we have completely addressed 6 out of these 8 points (W1,W3,Q1,Q3,Q4,Q5) and substantially addressed the remaining points (W2,Q2). Please let us know if there are any additional questions or if further clarification would be helpful. If the reviewer feels that their concerns have been adequately addressed, we would be grateful if they would consider updating their evaluation.
>
> > Dependence on large commercials models, no comparison to open-source smaller or distilled models presented.
>
> qqS8-W1: Dependence on large commercial models.
>
> Thank you for the comment. In the original submission, we had already included one open-source model (GPT-oss-120B) in our ablation study (Figure 4, lines 477-482). To more thoroughly address this concern, we have now expanded the analysis to include two additional open-source LLMs: Qwen-3 32B and Llama-3 8B-Instruct. F1 scores with MosaCD are reported below. We find that MosaCD is equivalent to GPT-5 with Qwen and slightly weaker with Llama, indicating that its effectiveness does not rely on large commercial models. We have updated Figure 4 (lines 477-482) to include these new results.
>
> | Model                     | Asia | Insurance | Hepar2 |
> | ------------------------- | ---- | --------- | ------ |
> | **Claude 3.5 Haiku**      | 0.93 | 0.68      | 0.67   |
> | **Claude 4 Sonnet**       | 0.93 | 0.87      | 0.70   |
> | **GPT-4o-mini (Default)** | 0.93 | 0.87      | 0.72   |
> | **GPT-5**                 | 0.93 | 0.85      | 0.72   |
> | **Llama-3-8B**            | 0.93 | 0.56      | 0.59   |
> | **GPT-OSS-120B**          | 0.67 | 0.72      | 0.68   |
> | **Qwen3-32B**             | 0.93 | 0.87      | 0.71   |
> *MosaCD F1 scores across LLMs*
>
> > The confidence-down propagation weights are heuristic; a probabilistic interpretation or adaptive β schedule would strengthen robustness.
>
> qqS8-W2: Confidence-down propagation weights.
>
> Thank you for raising this point. We agree that the confidence-down propagation step (Step 3) is heuristic, and that developing a probabilistic formulation could further strengthen robustness (we couldn’t find the exact definition of adaptive β schedule and interpreted it as a specific probabilistic formulation). Constructing such a principled variant is challenging in our setting because the joint distribution over nodes is unknown and difficult to estimate reliably. As a result, heuristic weighting strategies are commonly adopted in constraint-based methods. For example, Meek [1] and FCI [2] both must operate with a sequential strategy rather than jointly over all nodes . We have updated Section 2 (lines 87-88) to clarify this design choice and to highlight the development of a probabilistic framework as an important direction for future work.
>
> [1] Christopher Meek. Causal inference and causal explanation with background knowledge. arXiv preprint
>
> [2] Spirtes, Peter. "An anytime algorithm for causal inference." International Workshop on Artificial Intelligence and Statistics. PMLR, 2001.

---

> ### Author Response · Authors · 2025-11-24
> **Response to Reviewer qqs8 (2/3)**
>
> > Evaluation metrics relies on F1 of edge orientations only; additional causal sufficiency or equivalence-class metrics (e.g., SID, SHD) would provide a fuller view.
>
> qqS8-W3: Additional evaluation metrics.
>
> Thank you for this comment. We agree that including additional metrics provides a more comprehensive assessment. In response, we have added Structural Hamming Distance (SHD) to Appendix B. SHD leads to similar qualitative conclusions as the F1 metric: MosaCD continues to outperform the existing baselines in the same datasets. To further strengthen the evaluation, we also conducted new experiments with 8 additional baselines—4 LLM-based methods, 2 score-based methods, and 2 continuous-optimization methods. As shown in the updated table below, MosaCD achieves top performance in 7/10 datasets  and achieves the best overall performance. Together, these results offer a fuller view of the method’s performance, as requested. We have added this analysis to Section 6.1 line 377 and Appendix B.
>
> | Method             | Cancer (5) | Asia (8) | Child (20) | Insurance (27) | Water (32) |
> | ------------------ | ---------- | -------- | ---------- | -------------- | ---------- |
> | **PC**             | 2.0        | 3.0      | 10.5       | 28.0           | 53.0       |
> | **Meek**           | 2.0        | 3.0      | 6.0        | 19.0           | 38.0       |
> | **Shapley-PC**     | **0.0**    | 4.0      | 19.0       | 27.5           | 52.5       |
> | **ILS-CSL***       | 2.0        | **1.0**  | 5.5        | 25.0           | 37.5       |
> | **SCP***           | 2.0        | 3.0      | 6.0        | 21.0           | 38.0       |
> | **Jiralerspong***  | 3.0        | 5.0      | 38.0       | 52.0           | 71.0       |
> | **Vashishtha***    | 1.0        | **1.0**  | 9.0        | 21.0           | 34.0       |
> | **Causal Disco***  | 2.0        | 2.0      | 13.0       | 41.0           | **15.0**   |
> | **LLM-CD***        | **0.0**        | 2.0      | 30.0       | 59.0           | 81.0       |
> | **DAG-GNN**        | 4.0        | 9.0      | 23.0       | 60.0           | 55.0       |
> | **GES**            | 2.0        | 9.0      | 63.0       | 105.0          | 71.0       |
> | **HC**             | 2.0        | 9.0      | 51.0       | 95.0           | 60.0       |
> | **MosaCD*** (Ours) | **0.0**    | **1.0**  | **2.0**    | **12.0**       | 36.0       |
>
> | Method             | Mildew (35) | Alarm (37) | Hailfinder (56) | Hepar2 (70) | Win95pts (76) |
> | ------------------ | ----------- | ---------- | --------------- | ----------- | ------------- |
> | **PC**             | 23.0        | 12.0       | 105.0           | 99.5        | 68.0          |
> | **Meek**           | 17.0        | 7.5        | 97.0            | 87.0        | 57.5          |
> | **Shapley-PC**     | 18.5        | 14.0       | 105.0           | 89.0        | 69.5          |
> | **ILS-CSL***       | 10.0        | 12.0       | 96.5            | 77.5        | 54.0          |
> | **SCP***           | 17.0        | 8.0        | 99.0            | 88.0        | 56.0          |
> | **Jiralerspong***  | 50.0        | 80.0       | 72.0            | 137.0       | 135.0         |
> | **Vashishtha***    | 13.0        | 15.0       | 77.0            | 59.0        | 46.0          |
> | **Causal Disco***  | 24.0        | 27.0       | **53.0**        | **36.0**    | 69.0          |
> | **LLM-CD***        | 52.0        | 59.0       | 92.0            | 117.0       | 160.0         |
> | **DAG-GNN**        | 50.0        | 35.0       | 77.0            | 119.0       | 93.0          |
> | **GES**            | 152.0       | 101.0      | 150.0           | 122.0       | 122.0         |
> | **HC**             | 140.0       | 92.0       | 162.0           | 119.0       | 133.0         |
> | **MosaCD*** (Ours) | **8.0**     | **6.0**    | 90.0            | 52.0        | **37.0**      |
> *SHD scores across datasets and baselines*

---

> ### Author Response · Authors · 2025-11-24
> **Response to Reviewer qqS8 (3/3)**
>
> > What is the computational cost of the shuffle-and-vote procedure relative to the total CI test time?
>
> qqS8-Q1: Computational cost of the shuffle-and-vote procedure relative to the total CI test time.
>
> Thank you for the question. We have now measured and reported the computational cost of the shuffle-and-vote procedure and the total CI testing time across 3 datasets used in our ablation studies (“Asia,” “Insurance,” “Hepar2”) and 3 skeleton methods (“PC,” “PC-Stable,” “CPC”). Runtime in seconds is reported below and added to Appendix C. We find that the shuffle-and-vote step takes longer than the CI tests themselves (avg. ~10x  longer for Hepar2). Although this overhead is non-negligible, we find it acceptable in practice. Accessing a commercial model through their API may mitigate this, as parallel prompting would then be easily achievable. We have updated Section 6.1 (lines 421-422) and Appendix F to report these results. We acknowledge an efficient prompting method is an important future direction to maximize runtime efficiency.
>
>
> | Method          | Asia PC | Asia PC-Stable | Asia CPC | Insurance PC | Insurance PC-Stable | Insurance CPC | Hepar2 PC | Hepar2 PC-Stable | Hepar2 CPC |
> | --------------- | ------- | -------------- | -------- | ------------ | ------------------- | ------------- | --------- | ---------------- | ---------- |
> | **Skeleton**    | 0.23    | 0.50           | 0.55     | 50.32        | 76.51               | 75.89         | 187.05    | 313.21           | 264.50     |
> | **LLM**         | 256.82  | 232.13         | 264.14   | 1501.94      | 1397.03             | 1434.65       | 2492.51   | 2453.28          | 2260.18    |
> | **Propagation** | 0.00    | 0.00           | 0.00     | 0.01         | 0.01                | 0.01          | 0.03      | 0.03             | 0.03       |
> *Runtime (seconds) for each step across datasets and skeletons*
>
> > Could confidence propagation be formalized probabilistically (e.g., Bayesian edge belief updating)?
>
> qqS8-Q2: Probabilistic formulation of confidence propagation.
>
> Please see our response above to qqS8-W2.
>
> >  How does MosaCD handle contradictory LLM votes or inconsistent priors across iterations?
>
> qqS8-Q3: Handling contradictory LLM votes and inconsistent priors across iterations.
>
> Thank you for the question. The reviewer asked 2 related questions: (i) how MosaCD resolves conflicting LLM votes for the direction of a given edge, and (ii) how it handles LLM-proposed directions that are globally inconsistent (e.g., that would introduce cycles). We address each question in turn.
>
> (i) How MosaCD resolves conflicting LLM votes for the direction of a given edge.
> For each undirected pair X–Y, we query the LLM with 2 versions: one in the order (X → Y vs. Y → X) and one with the answer choices reversed. Each version is queried 5 times. We retain a direction only if it receives a strict majority of votes (at least 3/5) in both question orders. If the answers are inconsistent or fail to achieve a majority, we discard the edge direction.
>
> (ii) how it handles LLM-proposed directions that are globally inconsistent.
> Edges are processed in random order. If the LLM-proposed direction for X–Y contradicts the orientation of the partially constructed graph—such as introducing a directed cycle—it is discarded.
>
> We recognize that it is our responsibility to provide a clear exposition. We have updated Section 4 (line 205) in an effort to improve clarity.
>
> > Can the approach scale to 100–500 nodes or use learned embeddings to approximate CI filtering?
>
> qqS8-Q4: Scalability.
>
>
> Thank you for the comment. The reviewer raises 2 points, which we address in turn:
>
> Scaling to 100+ nodes: The method indeed works with larger graphs, running on the Andes dataset from bnlearn (223 nodes) runs in 2 hours and 15 minutes (2 hours for prompting, 15 minutes for skeleton searching). Please also refer to our response to qqS8-Q1 for how to improve computational efficiency for MosaCD.
>
> Approximate CI filtering: Approximate CI filtering can indeed improve skeleton search efficiency in conjunction with MosaCD, as it is agnostic to the skeleton construction procedure.
>
> We have updated Section 6.1 (lines 421-422) and Appendix F to report these findings and discuss scalability.
>
> > Can the method work with smaller or distilled language models that could be run locally?
>
> qqS8-Q5: Smaller or distilled language models.
>
> Thank you for the question. We conducted additional experiments using smaller, open-source models—specifically Qwen-3 32B and Llama-3 8B-Instruct—to evaluate whether MosaCD can operate effectively with locally runnable LLMs. We find that MosaCD achieves identical performance with Qwen-3 32B as it does with GPT-5, indicating that its effectiveness does not depend on large proprietary LLMs. Please see response to qqS8-W1 for more details.

---

### Author Response · Authors · 2025-12-02
**Final Overall Response**

Thank you for your efforts in handling submissions during the recent unexpected situation. To facilitate your evaluation, we provide a concise summary of the review history and our rebuttal progress.

During the rebuttal period, we conducted substantial new analyses, including:

(i) adding 8 new baseline methods (4 LLM-based methods, 2 score-based methods, 2 continuous-optimization methods), showing MosaCD outperforms all of these in 7/10 datasets and achieves the best overall performance by a wide margin;

(ii) new analysis assessing the impact of LLM memorization, and

(iii) new analysis assessing MosaCD’s ability to mitigate collider-first orientation issue.

The manuscript has been revised accordingly, incorporating 2 new analyses, 2 updated figures/tables, and 2 new or updated supplementary figures/tables.

Before the reviewer updates were reverted, our submission had an average score of 5. One reviewer (Mmpx) responded and increased their score from 2→4. Two additional reviewers (JtHM, g17M) explicitly stated they would increase their score conditional on satisfactory revisions—which we believe we have provided. We therefore expect that the final average score would have been significantly higher had reviewers been allowed to complete their updates.

Below we summarize our responses to each reviewer:

**Reviewer qqS8 (score 8).**

Raised 3 weaknesses and 5 questions. We believe that we have completely addressed 6 out of these 8 points (W1,W3,Q1,Q3,Q4,Q5) and substantially addressed the remaining points (W2,Q2). While we did not receive a response due to the shortened discussion period, we believe the reviewer would be satisfied with our rebuttal.

**Reviewer JtHM (score 4).**

Raised 8 comments/weaknesses/questions. We believe that we have completely addressed 2 out of these 3 points (W1,W3) and substantially addressed the remaining points (W2). The reviewer indicated: “I’ll be happy to increase my score if the authors tackle all points I mentioned.” (Official Review by Reviewer JtHM). As our revisions address these points, we would have expected an increased score.

**Reviewer g17M (score 4).**

Raised 3 comments/weaknesses/questions. We believe that we have completely addressed 7 out of these 8 points (W1,W2,W3,W5,W6,W7,W8) and substantially addressed the remaining points (W4). The reviewer explicitly stated: “I am happy to increase my score if the authors clarify I am wrong on the following comments.” (referring to that “method may fail to mitigate the collider-first issue”; Official Review by Reviewer g17M). We believe our clarification of the collider-first issue (response to g17M-W1) had completely addressed this concern.

**Reviewer Mmpx (score 2→4).** (“I updated my score accordingly” from latest reply on 27 Nov 2025 by Reviewer Mmpx)

Raised 8 comments/weaknesses/questions. We believe that we have completely addressed 7 out of these 8 points (W1,W2,W3,W5,W6,W7,W8) and substantially addressed the remaining points (W4). The reviewer replied twice during the discussion and left only one remaining concern regarding assumptions in related LLM-based methods, which we resolved in response to Mmpx-R2.

In summary, we believe we have substantially addressed all reviewer concerns, and multiple reviewers indicated willingness to raise their scores based on our revisions. We appreciate your consideration.

---

### Meta-Review · Area_Chair_6TNV · 2026-01-05

**Summary:**

The paper contributes (i) a theoretically grounded heuristics for causal discovery based on conditional independence tests (orienting non-colliders first), and empirically validates it, and (ii) MosaCD, a LLM and CI-based algorithm that leverages the new heuristics.
The first theoretical contribution is novel and significant, and it is likely to impact future works using conditional tests for causal discovery in the finite sample regime.
However, the insufficient evaluation of MosaCD and the misrepresentation of related works make the second contribution difficult to position (empirically and methodologically) w.r.t. the existing literature. Hence the current version of the paper is unsuitable for the conference.

**Reviewer Concerns:**

Two issues remain unsolved after rebuttal:
- **Evaluation on un-memorized datasets.** The evaluation is mainly carried out on datasets known to be memorized by LLMs. The revised paper now reports in the appendix two additional datasets that have been shown not to have this issue. Two baselines (classical PC methods) and F1 are reported. The authors rightfully highlight that MosaDC results, although much lower than on the originally considered datasets, are still better than the two considered baselines, indicating that MosaDC's improvements over classical PC methods is not exclusively due to memorization. However, the performance drop also indicates that MosaCD benefits from memorization to some degree. It would have been key to compare with the other LLM-based methods on these datasets, to assess whether MosaCD is more or less affected by memorization. Furthermore, performance should have been measured using the two complementary metrics, SHD and SID, that are standard in the causal discovery literature.
- **Misrepresentation of existing LLM-based methods.**
In the paper (and the rebuttal), the authors claim that "[LLM-based] methods implicitly assume that direct LLM outputs are reliable or have predictable error rate". However there exists at least one method (Long et al. 2023) that considers imperfect LLM outputs with unknown and non-constant error rate.

The following issues were sufficiently addressed: dependence on closed-source LLM, computational cost, missing baselines, sample complexity, too strong assumptions.

**Reviewer Scores:**

All issues raised by reviewer g17M were properly addressed. Most issues from reviewer Mmpx were also solved. Hence both their scores should have increased.

---

### Decision · Program_Chairs · 2026-01-26

Reject